# DATA PREDICTION DENOISING MODELS: THE PUPIL OUTDOES THE MASTER

## ABSTRACT

Due to their flexibility, scalability, and high quality, diffusion models (DMs) have become a fundamental stream of modern AIGC. However, a substantial performance deficit of DMs emerges when confronted with a scarcity of sampling steps. This limitation stems from the DM's acquisition of a series of weak denoisers obtained by minimizing a denoising auto-encoder objective. The weak denoisers lead to a decline in the quality of generated data samples in scenarios with few sampling steps. To address this, in this work, we introduce the Data-Prediction Denoising Model (DPDM), a constructor that embodies a sequence of stronger denoisers compared to conventional diffusion models. The DPDM is trained by initializing from a teacher DM. The core idea of training DPDM lies in improving the denoisers' data recovery ability with noisy data as inputs. We formulate such an idea through the minimization of suitable probability divergences between denoiser-recovered data distributions and the ground truth data distribution. The sampling algorithm of the DPDM is executed through an iterative process that interleaves data prediction and the sequential introduction of noise. We conduct a comprehensive evaluation of the DPDM on two tasks: data distribution recovery and the few-step image data generation. For the data distribution recovery, the DPDM shows significantly stronger ability to recover data distributions from noisy distribution. For the data generation task, we train DPDMs on two benchmark datasets: the CIFAR10, and the ImageNet$64 \times 64$. We compare the DPDM with baseline diffusion models together with other diffusion-based multi-step generative models under the few-step generation setting. We observe the superior performance advantage of DPDMs over competitor methods. In addition to the strong empirical performance, we also elucidate the interconnections and comparisons between the DPDM and existing methodologies, which shows that DPDM is a stand-alone generative model that is essentially different from existing models.

## 1 INTRODUCTION

In the past decade, the realm of deep generative models has marked substantial progress across diverse domains, encompassing data generation (Karras et al., 2020; 2022; Nichol and Dhariwal, 2021; Oord et al., 2016; Ho et al., 2022; Poole et al., 2022; Hoogeboom et al., 2022; Kim et al., 2022; Luo et al., 2023), density estimation (Kingma and Dhariwal, 2018; Chen et al., 2019), image editing (Meng et al., 2021; Couairon et al., 2022), and numerous others (Zhang et al., 2023; Yoon et al., 2021; Wang et al., 2022; Nie et al., 2022). Particularly notable, recent advancements in text-driven high-resolution image generation (Saharia et al., 2022; Ramesh et al., 2022; 2021; Rombach et al., 2022) have boldly extended the frontiers of employing generative models in the domain of Artificial Intelligence Generated Content (AIGC). Behind the empirical success are fruitful developments of a wide variety of deep generative models, among which diffusion models (DMs) (Ho et al., 2020; Song et al., 2020b) are the most prominent.

DMs leverage the diffusion processes and model the data across a wide spectrum of noise levels. Their ease of training, ability to scale, and high sample quality have made DMs the preferred option for generative modeling. There are two significant characteristics of diffusion models. First, a DM is a multi-step generative model, i.e., its generation process incorporates a composition of multiple evaluations of the model's network before outputting final samples. This multi-step character makes the DM flexible in practice and powerful in modeling, distinguishing it from previous models such as Generative Adversarial Networks (GANs) (Goodfellow et al., 2014; Radford et al., 2015; Brock et al., 2018; Karras et al., 2019; 2020); Second, the training of DMs can be interpreted as training

a series of denoising auto-encoders (Bengio et al., 2013; Vincent, 2011; Song et al., 2020b), making the concept of diffusion models easy to understand and aligned with human sense.

With the above two characteristics, DMs have shown promising generative performance with sufficient sampling steps with a sequence of *denoising* operations. In spite of its distinguished performance, the diffusion model has a significant drawback: *its performance drops significantly when the number of sampling steps is few, e.g. less than 10.* This means that DMs can not result in satisfactory samples when the computational resources are limited or there are strong demands on inference efficiency. For instance, sampling from DMs often requires up to a total number of 50+ evaluations of the deep neural network (NFEs) to give a promising performance, making it computationally inefficient. This disadvantage greatly limits the wider use of DMs, especially on devices with a limited computational ability such as mobile phones, and edge devices, or on other applications such as denoising-based adversarial defenses (Zhang et al., 2023; Nie et al., 2022).

This drawback of DMs strongly motivates us to understand the reason for the performance drop when the sampling steps are limited to be few. In this work, we reveal one of the most important reasons for such a performance deficit of DMs with limited NFEs through an empirical observation in Section 3.1 which shows:

> *The DM's training objective results in weak denoisers that have poor ability to recover the data distribution with noisy distributions, so the model can not give high-quality samples with few NFEs.*

Driven by the findings of weak denoisers, one practical way to improve the few-step generative performance is to enhance the capability of recovering data distribution. To this end, we introduce the **Data Prediction Denoising Model (DPDM)**, a stand-alone multi-step generative model that shows significantly stronger generative performance than DMs under a few sampling steps. The proposed DPDM is constructed by improving the data recovery ability of each data-prediction denoiser of a teacher DM by minimizing some well-defined distribution divergence. The enhanced denoisers are shown to be able to better recover data distributions from the observed noisy data distributions which the DMs can not achieve. With stronger denoisers, the DPDM significantly outperforms DMs with few NFEs.

To generate samples from DPDMs, we introduce a corresponding *Data Prediction Denosing Sampling* algorithm, which we call DPDM sampler for short. The DPDM sampler is a different sampling algorithm from the DMs' SDE or ODE sampler. It consists of a sequence of successive iterations of data-prediction denoising operations with denoisers and random noise additions. The DPDM sampler is essentially different from DMs in several aspects which we analyze in Section 5.2.

The proposed DPDM is essentially different from diffusion models in both training methods and sampling methods. The training method of diffusion models encounters an image reconstruction objective (a $L^2$ loss function) to train (potentially weak) denoisers. On the contrary, the DPDM is based on minimizing a novel smoothed KL divergence between the denoiser distribution and the ground truth data distribution. Each sampling step of DDPM gradually predicts the mean of the next-step denoised distribution. On the contrary, each step of DPDM's sampling step directly predicts clean data and then adds a Gaussian noise to obtain samples from the less-noisy distribution. Overall, the DPDM is a stand-alone multi-step generative model that differs from diffusion models in both concepts and implementations.

To demonstrate the effectiveness of DPDM, we apply DPDM on two tasks: recovering data distribution from noisy samples and the few-step image data generation. For the data distribution recovery experiment, we first add noise to ground truth clean data and then use the denoisers from DPDM and DM to denoise in order to recover the data distribution. The DPDM shows significantly better recovery performance than baseline diffusion models. For the few-step image data generation experiment, we train DPDM on two benchmark datasets: the CIFAR10, and the ImageNet$64 \times 64$ datasets, following the same settings as (Karras et al., 2022). For both datasets, the DPDM outperforms both the baseline DMs and other diffusion-based multi-step generative models, such as consistency models(Song et al., 2023), progressive distillation (Salimans and Ho, 2022), denoising diffusion GANs (Xiao et al., 2021), and others (Luo et al., 2023; Luhman and Luhman, 2021; Zheng et al., 2023; 2022; Liu et al., 2022; Zhao et al., 2023; Xue et al., 2023), under the few-step generation setting. Overall, the optimal performance of DPDM is comparable to the baseline DMs with about three times more sampling steps with DMs. All empirical experiments show strong performance of the proposed DPDM with limited sampling steps. Besides that, we also compare the DPDM with other multi-step generative models

in Section 4 in order to distinguish it from existing works. The comparison shows that the DPDM is an essentially different model than previous ones.

To make the presentation clear, we summarize our contribution as follows:

- We analyze the reason for the performance drop of DMs with few sampling steps, by showing that the DM's trained denoisers have limited ability to recover data distribution from noisy distributions; The analysis sheds some light on more improvements of DMs in the future;

- We introduce both the Data Prediction Denoising Model (DPDM), together with its training and sampling algorithms. We set a solid mathematical foundation for DPDM through divergence minimization;

- The experiments on image data generations demonstrate that DPDMs achieve the state-of-the-art few-step generation ability among diffusion-based multi-step generative models.

## 2 PRELIMINARY

Assume $p_0 = p_d$ denotes the underlying ground truth data distribution. For ease of mathematical exposition, we use $p_0$ and $p_d$ interchangeably. In generative modeling, we want to generate new samples $\boldsymbol{x}_0 \sim p_0(\boldsymbol{x}_0)$. Diffusion Models are the most potent generative models that use a neural network to approximate the marginal score functions of a forward diffusion process that is initialized with $p_0$.

**Diffusion models.** A diffusion model relies on a forward diffusion (2.1) to transform the initial distribution $p_0$ towards some simple noise distribution,

$$\mathrm{d}\boldsymbol{x}_t = \boldsymbol{F}(\boldsymbol{x}_t, t)\mathrm{d}t + G(t)\mathrm{d}\boldsymbol{w}_t, \tag{2.1}$$

where $\boldsymbol{F}$ is a pre-defined drift function, $G(t)$ is a pre-defined scalar-value diffusion coefficient, and $\boldsymbol{w}_t$ denotes an independent Wiener process. A multiple-level or continuous-indexed score network $\boldsymbol{s}_\phi(\boldsymbol{x}, t)$ is usually employed in order to approximate marginal score functions of the forward diffusion process (2.1). The learning of marginal score functions is achieved by minimizing a weighted denoising score matching objective (Vincent, 2011; Song et al., 2020b),

$$\mathcal{L}_{DSM}(\phi) = \int_{t=0}^T w(t)\mathbb{E}_{\boldsymbol{x}_0 \sim p_0, \boldsymbol{x}_t|\boldsymbol{x}_0 \sim p_t(\boldsymbol{x}_t|\boldsymbol{x}_0)}\|\boldsymbol{s}_\phi(\boldsymbol{x}_t, t) - \nabla_{\boldsymbol{x}_t}\log p_t(\boldsymbol{x}_t|\boldsymbol{x}_0)\|_2^2\mathrm{d}t. \tag{2.2}$$

Here the weighting function $w(t)$ controls the importance of the learning at different time levels and $p_t(\boldsymbol{x}_t|\boldsymbol{x}_0)$ denotes the conditional transition of the forward diffusion (2.1). In this paper, we only consider the simplest diffusion unless especially emphasizing, for which $\boldsymbol{x}_t = \boldsymbol{x}_0 + \sigma(t)\boldsymbol{\epsilon}$, $\boldsymbol{\epsilon} \sim \mathcal{N}(\boldsymbol{0}, \boldsymbol{I})$ where $\sigma(t)$ is a monotonic noise schedule function. Therefore, $\nabla_{\boldsymbol{x}_t}\log p_t(\boldsymbol{x}_t|\boldsymbol{x}_0) = -\frac{1}{\sigma(t)}\boldsymbol{\epsilon}$ and the training objective is simplified as (2.3):

$$\mathcal{L}_{DSM}(\phi) = \int_{t=0}^T w(t)\mathbb{E}_{\substack{\boldsymbol{x}_0 \sim p_0, \boldsymbol{\epsilon} \sim \mathcal{N}(\boldsymbol{0}, \boldsymbol{I}) \\ \boldsymbol{x}_t = \boldsymbol{x}_0 + \sigma(t)\boldsymbol{\epsilon}}}\left\|\boldsymbol{s}_\phi(\boldsymbol{x}_t, t) + \frac{1}{\sigma(t)}\boldsymbol{\epsilon}\right\|_2^2\mathrm{d}t. \tag{2.3}$$

The trained score network $\boldsymbol{s}_\phi(\boldsymbol{x}_t, t)$ is a good approximation of marginal score functions of diffused data distribution $\boldsymbol{s}_{p_t}(\boldsymbol{x}_t) := \nabla_{\boldsymbol{x}_t}\log p_t(\boldsymbol{x}_t)$, and high-quality samples can be drawn from DM by simulating reversed SDE which is implemented by learned score network (Song et al., 2020b). However, the simulation of an SDE is significantly slow and usually requires at least 20 sampling steps to obtain satisfied samples. Therefore, a promising line of work shows that distilling the pre-trained teacher diffusion model to an efficient student network is a strong method for enhancing generation efficiency.

## 3 DATA-PREDICTION DENOISING MODELS

The main goal of Data Prediction Denoising Models is to improve the data distribution recovery ability from noisy data distributions. To begin with, we revisit the training objective of the vanilla diffusion model from the noisy-denoising perspective and evaluate the effects of the trained denoiser model.

### 3.1 REVISIT THE TRAINING OF DIFFUSION MODEL AS DENOISERS OF NOISY DISTRIBUTION

**Revisit the Training of Diffusion Model.** Recall the diffusion model's training objective (2.3), when the conditional diffusion is a Gaussian noise addition, i.e., $\boldsymbol{x}_t|\boldsymbol{x}_0 \sim \mathcal{N}(\boldsymbol{x}_t; \boldsymbol{x}_0, \sigma^2(t)\boldsymbol{I})$, the noisy data writes $\boldsymbol{x}_t = \boldsymbol{x}_0 + \sigma(t)\boldsymbol{\epsilon}$, $\boldsymbol{\epsilon} \sim \mathcal{N}(\boldsymbol{0}, \boldsymbol{I})$, and the conditional score function writes $\nabla_{\boldsymbol{x}_t}\log p_t(\boldsymbol{x}_t|\boldsymbol{x}_0) = -(\boldsymbol{x}_t - \boldsymbol{x}_0)/\sigma^2(t)$, and the training objective (2.3) is equivalent to

$$\mathcal{L}_{Data-Pred}(\phi) = \int_{t=0}^{T} \frac{w(t)}{\sigma^4(t)} \mathbb{E}_{\boldsymbol{x}_0 \sim p_0, \boldsymbol{x}_t|\boldsymbol{x}_0 \sim p_t(\boldsymbol{x}_t|\boldsymbol{x}_0)} \|\boldsymbol{d}_\phi(\boldsymbol{x}_t, t) - \boldsymbol{x}_0\|_2^2 \mathrm{d}t. \tag{3.1}$$

The equation indicates that the map $\boldsymbol{d}_\phi(\boldsymbol{x}_t, t) := \boldsymbol{x}_t + \sigma^2(t)\mathbf{s}_\phi(\boldsymbol{x}_t, t)$ is trained to predict the clean data $\boldsymbol{x}_0$ with the noisy data $\boldsymbol{x}_t = \boldsymbol{x}_0 + \sigma(t)\boldsymbol{\epsilon}$. This objective form is called the denoising objective and the trained model $\boldsymbol{d}_\phi(\boldsymbol{x}_t, t)$ is called a denoising auto-encoder (DAE) (Bengio et al., 2013; Vincent, 2011). To some extent, the trained $\boldsymbol{d}_\phi$ is supposed to be able to recover clean data $\boldsymbol{x}_0$ based on noisy data $\boldsymbol{x}_t$, therefore there is an intuition that $\boldsymbol{d}_\phi$ should be able to recover clean data distribution $p_0$ from noisy data distribution $p_t$.

**DM's Denoisers are Weak for Recovering Data Distributions.** The optimal denoiser trained with (3.1) has an explicit form $\boldsymbol{d}^*(\boldsymbol{x}_t, t) = \mathbb{E}[\boldsymbol{x}_0|\boldsymbol{x}_t]$, such an denoiser tends to predict an *average* denoised sample based on current noisy data $\boldsymbol{x}_t$. When the added noise of $\boldsymbol{x}_t$ is significantly large, the *averaged denoised data* is blurred, which prevents the diffusion model from performing well with few generation steps. To numerically verify the ability of the diffusion model's denoisers to recover the data distribution from noisy distributions, as demonstrated in Figure A.4, we take a real image of a resolution of $64 \times 64$ (i.e. a lovely dog in the Figure) and add different amount of Gaussian noise to it to obtain noisy data. Then we use the pre-trained class conditional diffusion model from Karras et al. (2022). We use the diffusion model's corresponding denoisers to denoise noisy data and observe the denoised data sample. We find that when the variance of added Gaussian noise grows, the diffusion model's denoisers become unable to produce high-quality samples based on noisy inputs. This indicates that using denoisers of diffusion models to directly generate clean data from noisy data (i.e. random Gaussian noise) will lead to poor performance. This weak ability of denoisers prevents the diffusion model from generating high-quality samples when the sampling steps are set to be small. This finding motivates us to propose algorithms that can enhance the ability to recover clean data distribution from noisy ones.

### 3.2 ENHANCING THE DENOISER THROUGH VARIATIONAL INFERENCE

**Enhancing a single denoiser.** Our goal is to obtain a denoiser that can exactly recover the clean data distribution. We begin by introducing the basic setting. Let $\boldsymbol{x}_0 \sim p_0$ be a clean data sample. Without loss of generality, we consider the DM's denoiser function for a fixed noise sigma $\sigma(t) = 1$ to make the discussion clear. Therefore, the noisy sample $\boldsymbol{x}_t = \boldsymbol{x}_0 + \boldsymbol{\epsilon}$, $\boldsymbol{\epsilon} \sim \mathcal{N}(\boldsymbol{0}, \boldsymbol{I})$. Let $\boldsymbol{d}(\boldsymbol{x}_t, t)$ denote a denoiser that is supposed to be able to recover clean data distribution with noisy data as inputs for a fixed noise variance $\sigma(t) = 1$. We omit the second argument $t$ and use $\boldsymbol{d}(\boldsymbol{x}_t)$ for short. Let $p_t(\boldsymbol{x}_t)$ denote the noisy distribution, and $q_0(\widehat{\boldsymbol{x}}_0)$ denotes the denoised distribution: $\widehat{\boldsymbol{x}}_0 \sim q_0(\widehat{\boldsymbol{x}}_0)$, $\widehat{\boldsymbol{x}}_0 = \boldsymbol{d}_\theta(\boldsymbol{x}_t)$. Let $q_t(\widehat{\boldsymbol{x}}_t)$ denote the noisy denoised distribution, for which another independent Gaussian noise is added to the denoised data with $\widehat{\boldsymbol{x}}_t = \widehat{\boldsymbol{x}}_0 + \boldsymbol{\epsilon}$.

The goal is to minimize some probability divergence between $q_0$ and $p_0$, such that the denoised distribution $q_0$ can closely match the clean data distribution $p_0$, therefore the denoiser obtains a strong data distribution recovery ability. More precisely, we consider the *smoothed KL divergence*, which is defined as:

$$\mathcal{D}_{smooth-KL}(q_0, p_0) := \mathcal{D}_{KL}(q_t, p_t) = \mathbb{E}_{\boldsymbol{x}_t \sim q_t} \log \frac{q_t(\boldsymbol{x}_t)}{p_t(\boldsymbol{x}_t)} \tag{3.2}$$

Here $p_t(\boldsymbol{x}_t)$ denote the distribution of $\boldsymbol{x}_t = \boldsymbol{x}_0 + \boldsymbol{\epsilon}, \boldsymbol{x}_0 \sim p_0(\boldsymbol{x}_0), \boldsymbol{\epsilon} \sim \mathcal{N}(\boldsymbol{0}, \boldsymbol{I})$, and $q_t(\widehat{\boldsymbol{x}}_t)$ is defined as the distribution of $\widehat{\boldsymbol{x}}_t = \boldsymbol{d}_\theta(\boldsymbol{x}_t) + \epsilon, \boldsymbol{x}_t \sim p_t(\boldsymbol{x}_t), \boldsymbol{\epsilon} \sim \mathcal{N}(\boldsymbol{0}, \boldsymbol{I})$. From a high-level concept, the distribution $p_t$ and $q_t$ stands for the noisy and noisy-denoised distribution, respectively. It is easy to check that the smoothed KL divergence is a well-defined divergence and satisfies some robustness for distributions with misaligned density support (check Appendix A.1 for details).

**Proposition 3.1.** The $\mathcal{D}_{smooth-KL}(q_0, p_0)$ satisfies that $\mathcal{D}_{smooth-KL}(q, p) \geq 0, \forall q, p$. Furthermore, the equality holds if and only if $q = p$, almost everywhere under measure $q$.

---

**Algorithm 1:** Training Algorithm of DPDM

---

**Input:** pre-trained DM $\boldsymbol{s}_{p_t}(.)$, multi-step denoiser model $\boldsymbol{d}_\theta(.,.)$, data samples $\boldsymbol{x}_0 \sim p_0$,
auxiliary diffusion model $\boldsymbol{s}_\phi(.,.)$; variance function of diffusion noise schedule $\sigma(t)$.

**while** *not converge* **do**

Update auxiliary diffusion model $\boldsymbol{s}_\phi$ using SGD with gradient

$$\mathrm{Grad}(\phi) = \frac{\partial}{\partial\phi}\int_{t=0}^{T} w(t)\mathbb{E}_{\substack{\boldsymbol{x}_0\sim p_0, \boldsymbol{\epsilon},\tilde{\boldsymbol{\epsilon}}\sim\mathcal{N}(\boldsymbol{0},\boldsymbol{I}),\\ \boldsymbol{x}_t=\boldsymbol{x}_0+\sigma(t)\boldsymbol{\epsilon},\tilde{\boldsymbol{x}}_0=\boldsymbol{d}_\theta(\boldsymbol{x}_t),\tilde{\boldsymbol{x}}_t=\tilde{\boldsymbol{x}}_0+\sigma(t)\tilde{\boldsymbol{\epsilon}}}} \left\|\boldsymbol{s}_\phi(\tilde{\boldsymbol{x}}_t,t)+\frac{1}{\sigma(t)}\tilde{\boldsymbol{\epsilon}}\right\|_2^2 \mathrm{d}t.$$

Update $\theta$ using SGD with the gradient

$$\mathrm{Grad}(\theta) = \int_{t=0}^{T} w(t)\mathbb{E}_{\substack{\boldsymbol{x}_0\sim p_0, \boldsymbol{\epsilon},\tilde{\boldsymbol{\epsilon}}\sim\mathcal{N}(\boldsymbol{0},\boldsymbol{I}),\\ \boldsymbol{x}_t=\boldsymbol{x}_0+\sigma(t)\boldsymbol{\epsilon},\tilde{\boldsymbol{x}}_0=\boldsymbol{d}_\theta(\boldsymbol{x}_t),\tilde{\boldsymbol{x}}_t=\tilde{\boldsymbol{x}}_0+\sigma(t)\tilde{\boldsymbol{\epsilon}}}} \left[\boldsymbol{s}_\phi(\tilde{\boldsymbol{x}}_t,t)-\boldsymbol{s}_{p_t}(\tilde{\boldsymbol{x}}_t)\right]\frac{\partial\tilde{\boldsymbol{x}}_t}{\partial\theta}\mathrm{d}t.$$

**end**

**return** $\theta,\phi$.

---

**Algorithm 2:** Sampling Algorithm of DPDM

---

**Input:** trained DPDM $\boldsymbol{d}_\theta(.,.)$, noise schedules $\{\sigma(t_k)\}_{k=1,\dots,K}$, initial samples $\boldsymbol{x}_0=\boldsymbol{0}$

**for** *k from* $1$ *to* $K$ **do**

Add noise: $\boldsymbol{x}_t=\boldsymbol{x}_0+\sigma(t_k)\boldsymbol{\epsilon},\boldsymbol{\epsilon}\sim\mathcal{N}(\boldsymbol{0},\boldsymbol{I})$;

Denoise: $\boldsymbol{x}_0=\boldsymbol{d}_\theta(\boldsymbol{x}_t,t_k)$

**end**

**return** $\boldsymbol{x}_0$.

---

In order to minimize the smoothed KL divergence (3.2) to learn a strong denoiser that is capable of recovering clean data distribution, we need to know how to update the denoiser's parameter. So in Theorem 3.2, we give an explicit gradient formula for minimizing the smoothed KL divergence.

**Theorem 3.2.** Let $\boldsymbol{s}_{q_t}(\boldsymbol{x}_t) := \nabla_{\boldsymbol{x}_t}\log q_t(\boldsymbol{x}_t)$ and $\boldsymbol{s}_{p_t}(\boldsymbol{x}_t) := \nabla_{\boldsymbol{x}_t}\log p_t(\boldsymbol{x}_t)$, the gradient of the smoothed KL in (3.2) between $q_0$ and $p_0$ is

$$\mathrm{Grad}(\theta) = \mathbb{E}_{\substack{\boldsymbol{x}_t=\boldsymbol{d}_\theta(\boldsymbol{x}_0+\sigma(t)\boldsymbol{\epsilon})+\sigma(t)\tilde{\boldsymbol{\epsilon}}\\ \boldsymbol{x}_0\sim p_0, \boldsymbol{\epsilon},\tilde{\boldsymbol{\epsilon}}\sim\mathcal{N}(\boldsymbol{0},\boldsymbol{I})}} \left[\boldsymbol{s}_{q_t}(\boldsymbol{x}_t)-\boldsymbol{s}_{p_t}(\boldsymbol{x}_t)\right]\frac{\partial\boldsymbol{x}_t}{\partial\theta}. \tag{3.3}$$

We put the proof of Theorem 3.2 in Appendix A.2. In gradient formula (3.3), we do not know the score functions $\boldsymbol{s}_{q_t}(\cdot)$ and $\boldsymbol{s}_{p_t}(\cdot)$. However, in practice, with the improvements in the design of score networks (Peebles and Xie, 2022; Gao et al., 2023; Wei et al., 2023) in diffusion models, and the improved techniques of score matching (Song et al., 2019; Vincent, 2011), the score function $\boldsymbol{s}_{p_t}(.)$ of noisy data distribution $p_t$ can be approximated with high accuracy by a well-pre-trained teacher diffusion model's score function (we may use $\boldsymbol{s}_{p_t}(\cdot)$ to replace the pre-trained diffusion model interchangeably). The unknown denoiser's score function $\boldsymbol{s}_{q_t}(\cdot)$ can be approximated by another auxiliary diffusion model $\boldsymbol{s}_\phi(\boldsymbol{x}_t,t)$ which is fine-tuned with denoised data $\tilde{\boldsymbol{x}}_0$. Therefore, by alternating the training iterations between fine-tuning $\boldsymbol{s}_\phi(.)$ and updating denoiser $\boldsymbol{d}_\theta$ with gradient (3.3), the denoiser will be trained to minimize the smoothed KL divergence, therefore match the denoising distribution and the clean data distribution.

**Generalize to multiple denoising levels.** In previous paragraphs, we have derived the training method for a single denoiser through the formulation of smoothed KL divergence minimization. However, in practice, we want a *multi-step* generative model other than a single denoiser, so that the performance and efficiency of data generation can be traded off. Therefore, in this paragraph, we generalize the training method for single denoisers to multiple denoisers (continuously time-indexed denoisers). To this end, we assume the DPDM model is parametrized as $\{\boldsymbol{d}_\theta(\boldsymbol{x}_t,t)\}_{t\in[0,T]}$ where for each time index $t$, the mapping $\boldsymbol{d}_\theta(\boldsymbol{x}_t,t)$ accounts for a denoiser that aims to recover the data distribution $p_0$ with noisy distribution $p_t$ as the input. To make this, we generalize the gradient formula

(3.3) for the single denoiser to an integral version (3.4):

$$\text{Grad}(\theta) = \int_0^T w(t)\mathbb{E}_{\substack{\widetilde{\boldsymbol{x}}_t = \boldsymbol{d}_\theta(\boldsymbol{x}_0 + \sigma(t)\boldsymbol{\epsilon}) + \sigma(t)\bar{\boldsymbol{\epsilon}} \\ \boldsymbol{x}_0 \sim p_0, \boldsymbol{\epsilon}, \bar{\boldsymbol{\epsilon}} \sim \mathcal{N}(\boldsymbol{0}, \boldsymbol{I})}} \left[ \boldsymbol{s}_{q_t}(\widetilde{\boldsymbol{x}}_t, t) - \boldsymbol{s}_{p_t}(\widetilde{\boldsymbol{x}}_t, t) \right] \frac{\partial \widetilde{\boldsymbol{x}}_t}{\partial \theta} dt. \tag{3.4}$$

Here the weighting function $w(t)$ controls the importance of learning each denoisers. In practice, we find that using the same weighting function as diffusion training (2.3) for gradient formula (3.4) works well. The objective integrates the training of all denoisers by considering each noise variance level in one integral. Besides, the score functions $\boldsymbol{s}_{q_t}(\widetilde{\boldsymbol{x}}_t, t)$ for each denoiser can be learned by a unified diffusion model $\boldsymbol{s}_\phi(\widetilde{\boldsymbol{x}}_t, t)$, for which the data with each $t$ is generated by adding noise to denoised samples, i.e. $\widetilde{\boldsymbol{x}}_t = \widetilde{\boldsymbol{x}}_0 + \sigma(t)\boldsymbol{\epsilon}, \boldsymbol{\epsilon} \sim \mathcal{N}(\boldsymbol{0}, \boldsymbol{I})$. Overall, the training algorithms of DPDM can be summarized in Algorithm 1.

**Sampling Algorithms.** With the training algorithm 1, strong denoisers (i.e. the DPDM) can be trained which are capable of recovering data distribution with noisy samples. To make use of these denoisers for sampling, we introduce a novel sampling algorithm 2 for DPDM. The algorithm 2 consists of multiple iterations of denoising and noise addition operations at a series of monotonically decreasing noise levels $\{\sigma(t_k)\}_{k=1,\dots,K}$. In the initial sampling step, the denoiser takes a pure random noise with a sufficiently large variance $\sigma(t_1)$ and outputs a predicted data sample. Then a Gaussian noise with variance $\sigma(t_2)$ is added to the output data. Then the iteration loops from $k=1$ to $k=K$ and outputs the final generated sample. Figure A.4 gives a comprehensive demonstration of how the DMDM's sampling algorithms work. The alternative of the data-prediction denoising and Gaussian noise addition steps gradually transforms an initial random noise into high-quality generated data. The DPDM sampling algorithm is essentially different from the sampling algorithms of diffusion models. One of the most significant differences lies in the continuity of the sampling trajectory. The diffusion model's sampling relies on the simulation of an SDE or ODE, which has strictly continuous trajectories. On the contrary, the sampling of DPDM relies on data-prediction denoising and adding a large independent Gaussian noise, therefore, its trajectory is not continuous.

## 4 RELATED WORK

**Comparison with Diffusion Models.** The diffusion model is a representative multi-step generative model. As we analyze in section 2, the diffusion model minimizes the denoising score matching objective (2.2) to train a series of neural-parametrized denoisers. After training, generated samples are obtained by numerically simulating generative SDEs or ODEs (Xue et al., 2023; Zhao et al., 2023; Lu et al., 2022; Karras et al., 2022; Song et al., 2020b) which is implemented with the trained neural score functions (i.e. denoisers). Though diffusion and DPDM share some common concepts, such as denoisers and multi-step generations, they are fundamentally different. First, as we show empirically in section 3.1, the denoisers of DM have poor data distribution recovery ability when the scale added noise is relatively large. On the contrary, as we will show in Section 5.1, the denoisers of DPDM are shown to be able to better recover the clean data distribution, which indicates that the DPDMs have strictly stronger denoisers than DMs. Second, the sampling algorithms of DPDMs and DMs are very different. The sampling algorithm of DPDM consists of multiple iterations of alternatives for denoising and noise additions whose trajectory is not continuous. On the contrary, the sampling algorithms of DMs solve a numerical ODE or SDE whose trajectory is continuous.

**Comparison with Consistency Distillation.** The consistency model (Song et al., 2023) (CM) is another representative multi-step generative model. It is trained by minimizing the self-consistency function of the generative ODE of a teacher DM. After training, sampling from CM can be obtained by iterations of denoising and noise addition, which is conceptually the same as the DPDM sampling. However, the DPDM and CM have many differences. First, the training of CM (through distillation) requires the simulation of the generative ODE, however, on the contrary, the training of DPDM does not require numerically simulating an ODE. Second, as is introduced in Song et al. (2023) the CM requires both a learned data metric (i.e. the LPIPS (Zhang et al., 2018). Such a learned neural metric is tricky in practice and usually unavailable. On the contrary, the DPDM does not require any learned neural metric, making it user-friendly in practice. Due to the page limitations, we put more discussions on related works in Appendix A.3.

Table 1: Comparison of clean data recovery ability of DPDM and baseline EDM diffusion model's denoisers on CIFAR10 dataset with a different standard deviation of added Gaussian noises. The quality of clean data distribution recovery is measured by FID, which is the lower the better.

| FID↓/$\sigma$ of Noise | 0.02 | 0.05 | 0.1 | 0.2 | 0.5 | 0.75 | 1.0 | 2.0 | 5.0 | 7.5 | 10.0 |
|---|---|---|---|---|---|---|---|---|---|---|---|
| DM (EDM) | 0.40 | 1.94 | 4.28 | 7.90 | 15.77 | 21.66 | 28.46 | 61.94 | 161.63 | 205.44 | 229.71 |
| **DPDM(ours)** | **0.13** | **0.37** | **0.65** | **1.08** | **2.46** | **4.40** | **7.11** | **23.59** | **100.86** | **155.98** | **192.47** |

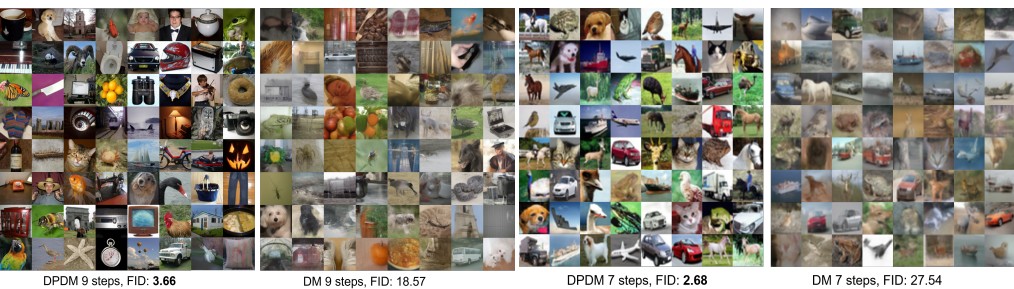

DPDM 9 steps, FID: **3.66**    DM 9 steps, FID: 18.57    DPDM 7 steps, FID: **2.68**    DM 7 steps, FID: 27.54

Figure 1: Comparison samples from DPDM and DM under few-step generation. *Left Two*: ImageNet-$64 \times 64$ (conditional); *Right Two*: CIFAR-10 (unconditional).

## 5  EXPERIMENTS

The core concept of DPDM is the strong data distribution recovery ability of denoisers over diffusion models. In this section, we choose the state-of-the-art diffusion model from the seminal work by Karras et al. (2022) (we denote the EDM model) as the baseline diffusion model and consider training DPDMs with the same architecture and diffusion process.

To quantitatively verify the data distribution recovery ability, in Section 5.1, we conduct an experiment on DPDM and baseline DMs to recover data distributions from noisy data distributions at different noise variances. The result reveals that the DPDM's data distribution recovery ability is consistently stronger than baseline DMs, which supports our claim and motivations in Section 3.1.

In Section 5.2, we use the trained DPDM for few-step image data generation. We evaluate the sampling performances of DPDM and baseline EDM diffusion models with limited NFEs (less than 10 NFEs) on two datasets. We measure the generative performances in terms of the Fretchat Inception Score (FID (Heusel et al., 2017), the lower the better), which is a widely used metric for evaluating generative modeling performances. On both datasets, the DPDM outperforms baseline DMs, together with other diffusion-based multi-step generative models, significantly under the few NFE settings.

### 5.1  QUATITATIVE MEASUREMENT OF DISTRIBUTION RECOVERY ABILITY

**Experiment Settings.**   In this experiment, we choose the CIFAR10 dataset to compare the data distribution recovery ability of DPDM and DMs across different noise variances. The standard deviation of added Gaussian noises is set to $0.02, 0.05, 0.1, 0.2, 0.5, 0.75, 1.0, 2.0, 5.0, 7.5, 10.0$ which varies from both small and large values. The DM and the DPDM are trained on the CIFAR10 dataset. We add Gaussian noises with different variances to the training data to obtain noisy data, and then input the noisy data into the corresponding denoiser of DPDM and DM to output recovered data. To quantitatively measure the recovery ability of the data distribution. We compute the FID values between two sets of clean data (i.e. the ground truth training data) and the recovered data. The FID values account for how similar the two sets of images are in the sense of distributional matching. The smaller the FID value is, the better distribution recovery the denoiser shows.

**Performances.**   Table 1 summarizes the recovery FID of DPDM and DM. On all variance levels, the DPDM's induced denoisers consistently outperform the baseline diffusion models by a significant margin. In Figure A.4, we provide a demonstration of how DPDM and the DM recover the data distribution from noisy samples. We visualize the original data, the noisy data, and the recovered data

Table 2: Unconditional sample quality on CIFAR-10 through multi-step models.

| METHOD | NFE ($\downarrow$) | FID ($\downarrow$) |
|---|---|---|
| **Sufficient Steps (more than 10)** | | |
| DDPM (Ho et al., 2020) | 1000 | 3.17 |
| LSGM (Vahdat et al., 2021) | 147 | 2.10 |
| PFGM (Xu et al., 2022) | 110 | 2.35 |
| EDM (Karras et al., 2022) | 35 | **1.97** |
| DDIM (Song et al., 2020a) | 15 | 12.23 |
| DPM-solver (Lu et al., 2022) | 15 | 5.13 |
| UniPC (Zhao et al., 2023) | 15 | 5.02 |
| EDM-ODE (Karras et al., 2022) | 15 | 5.62 |
| EDM-SDE (Karras et al., 2022) | 15 | 8.94 |
| SA-Solver (Xue et al., 2023) | 15 | 4.91 |
| **Few Step (less than 10)** | | |
| EDM (Karras et al., 2022) | 10 | 15.5 |
| 3-DEIS (Zhang and Chen, 2022) | 10 | 4.17 |
| UniPC (Zhao et al., 2023) | 8 | 5.10 |
| PD (Salimans and Ho, 2022) | 8 | 3.92 |
| CD (Song et al., 2023) | 8 | 2.92 |
| UniPC (Zhao et al., 2023) | 5 | 23.22 |
| Denoising-Diffusion GAN(T=2) (Xiao et al., 2021) | 2 | 4.08 |
| PD (Salimans and Ho, 2022) | 2 | 5.13 |
| CT (Song et al., 2023) | 2 | 5.83 |
| CD (Song et al., 2023) | 2 | 2.93 |
| Denoising-Diffusion GAN(T=1) (Xiao et al., 2021) | 1 | 14.6 |
| DFNO (Zheng et al., 2022) | 1 | 4.12 |
| CT (Song et al., 2023) | 1 | 8.70 |
| 1-ReFlow (+distill) (Liu et al., 2022) | 1 | 6.18 |
| 2-ReFlow (+distill) (Liu et al., 2022) | 1 | 4.85 |
| 3-ReFlow (+distill) (Liu et al., 2022) | 1 | 5.21 |
| PD (Salimans and Ho, 2022) | 1 | 8.34 |
| CD (Song et al., 2023) | 1 | 3.55 |
| Diff-Instruct (Luo et al., 2023) | 1 | 4.53 |
| **DPDM (Ours)** | 5 | 3.85 |
| **DPDM (Ours)** | 8 | **2.68** |

Table 3: Class-conditional sample quality ImageNet $64 \times 64$ through multi-step models.

| METHOD | NFE ($\downarrow$) | FID ($\downarrow$) |
|---|---|---|
| **Sufficient Steps (more than 10)** | | |
| ADM (Dhariwal and Nichol, 2021) | 250 | **2.07** |
| SN-DDIM (Bao et al., 2022) | 100 | 17.53 |
| EDM-SDE (Karras et al., 2022) | 79 | 2.44 |
| EDM-SDE (Karras et al., 2022) | 25 | 4.26 |
| GGDM (Watson et al., 2022) | 25 | 18.4 |
| EDM-SDE (Karras et al., 2022) | 15 | 8.94 |
| EDM-ODE (Karras et al., 2022) | 15 | 4.78 |
| SA-Solver (Xue et al., 2023) | 15 | 3.41 |
| UniPC (Zhao et al., 2023) | 15 | 3.41 |
| DPM-Solver (Lu et al., 2022) | 15 | 3.49 |
| **Few Step (less than 10)** | | |
| EDM-ODE(Karras et al., 2022) | 10 | 24.95 |
| EDM-ODE(Karras et al., 2022) | 8 | 33.90 |
| PD (Salimans and Ho, 2022) | 8 | 5.22 |
| CD (Song et al., 2023) | 8 | 4.70 |
| CT (Song et al., 2023) | 2 | 11.1 |
| PD (Salimans and Ho, 2022) | 2 | 8.95 |
| CD (Song et al., 2023) | 2 | 4.70 |
| PD (Salimans and Ho, 2022) | 1 | 15.39 |
| CT (Song et al., 2023) | 1 | 13.00 |
| CD (Song et al., 2023) | 1 | 6.20 |
| Diff-Instruct (Luo et al., 2023) | 1 | 5.57 |
| **DPDM (Ours)** | 5 | 3.98 |
| **DPDM (Ours)** | 8 | **3.66** |

by DPDM and DM respectively. The result shows that the DPDM is able to recover high-quality clean data from noisy samples, while the DM can only recover the rough shape of the original data.

## 5.2 FEW-STEP IMAGE GENERATION

**Settings.** In this experiment, we show the superior generative performance of the DPDMs under a few sampling steps. we train and use the DPDM on image generation on the CIFAR-10 (Krizhevsky et al., 2014) and the ImageNet $64 \times 64$ (Deng et al., 2009) datasets. We compare the DPDM with the baseline EDM (Karras et al., 2022) models and other diffusion-based multi-step generative models. We evaluate the performance of the trained DPDM via FID. More experiment details are put in Appendix B.1.

**Performances.** Tables 2 and 3 summarize the FID values of DPDM, the baseline EDMs, along with other multi-step generative models on the CIFAR10 datasets (unconditional without labels) and the conditional generation on the ImageNet $64 \times 64$ data. The notation NFE refers to sampling steps. On both datasets, DPDM performs significantly better than both baseline EDMs and other diffusion-based multi-step models under the few NFE settings (NFE no larger than 10).

As shown in Table 3 and 2, DPDM shows an FID of $3.66$ on ImageNet$64 \times 64$ with only 8 NFEs and an FID of $2.68$ on CIFAR10 unconditional generation with 8 NFEs. On both datasets, DPDM shows the best FID value among both diffusion-based few-step models and few-step diffusion solvers, such as consistency distillation (CD) and the consistency training (Song et al., 2023), the progressive distillation (Salimans and Ho, 2022), the Diff-Instruct (Luo et al., 2023), the DD-GAN (Xiao et al., 2021) and the diffusion model with various sampling algorithms such as SA-SolverXue et al. (2023), UniPC(Zhao et al., 2023), and the DPM-Solver (Lu et al., 2022). It is worth noting that the CD, CT, Diff-Instruct, PD, and EDM with SDE/ODE samplers all share the same model architecture and the teacher diffusion model (if necessary), therefore the comparison with these models especially highlights the promising few-step generation ability of DPDM. Besides, According to Table 2, the 5-step generation performance of DPDM (with an FID of 3.85) is better than the 15-step generation performance of the baseline EDM model (with an FID of 5.62). This indicates that DPDM achieves about 3 times more efficiency than baseline diffusion models with competitive performances.

Table 4: Comparison of FID values of DPDM and baseline EDM model on CIFAR10 and ImageNet64 datasets. The DPDM shows significantly better FID than the baseline Diffusion Model with few NFEs.

| FID↓/NFE | 4 | 5 | 6 | 7 | 8 | 9 | 10 | 12 | 14 | 16 | 18 |
|---|---|---|---|---|---|---|---|---|---|---|---|
| EDM | 95.25 | 68.71 | 51.07 | 36.59 | 28.69 | 22.97 | 18.98 | 13.92 | 7.14 | 4.52 | 3.24 |
| **DPDM(ours)** | **7.16** | **3.85** | **2.77** | **2.72** | **2.68** | **2.68** | **2.68** | **2.68** | **2.68** | **2.68** | **2.68** |

Figure 1 shows some non-cherry-picked few-step generated samples from DPDM and baseline EDM models on the ImageNet$64 \times 64$ and the CIFAR10 datasets. The DPDM data is generated with sampling algorithms 2, while the DM samples are generated with DM's default ODE samplers. We find that using DPDM's sampling algorithm for DM leads to worse performance than DM's original sampling algorithms. As is shown in Figure 1, the DPDM is able to generate high-quality samples with good diversities with no more than 10 NFEs. On the contrary, the diffusion model fails to generate satisfactory samples with such few sampling steps. In conclusion, the DPDM achieves promising few-step generative performance under challenging few-step settings. We put more discussions and analyses in the Appendix B.1.

**Detailed Comparison with Different Sampling Steps.** Table 4 shows the detailed FID values of DPDM and baseline EDM models on the CIFAR10 dataset with different NFEs. The result shows that the DPDMs show significantly superior performance than EDM with limited NFEs (i.e. 4-18 NFEs). Such an advantage arises from the improved denoisers of DPDM over baseline DMs. As is shown in Table 4, the FID value of DPDM quickly becomes less than 3.0 with 6 NFEs, and the optimal FID is attained with 9 NFEs. After that, the FID value of DPDM saturates and holds about 2.68, which is also better than the baseline EDM model with 18 NFEs. We also give a Figure A.4 which visualizes batches of samples from the same DPDM model on CIFAR10 with different sampling steps. The visualization shows that the generated samples become poor when the NFEs are set to be less than 5.

**Exploration the Combination of Model and Sampling Algorithms.** In this paragraph, we give an intuitive exploration of the DPDM's sampling algorithm 2 with the diffusion model's default sampling algorithms. Since both DPDM and DM can provide denoisers, therefore the combination of model and sampling algorithms seems flexible. To explore the effects of using DPDM and DM with different sampling algorithms, we try the four sampling configurations on the CIFAR10 unconditional generation task, which are DPDM+DPDM sampler, DM+DPDM sampler, DPDM+Diffusion sampler, and DM+diffusion sampler. We calculate the FID as a metric and also visualize the generated samples with 7 NFEs in Figure A.4. As is shown in Figure A.4, the DPDM with DPDM sampler shows the best performances, and the DM with DM sampler shows the second. The cross-use of models and samplers gives very low-quality samples. This result shows that the DPDM model and sampler are closely paired, which demonstrates from another aspect that the DPDM is totally different from diffusion models. We think more exploration and development of more advanced DPDM samplers is an important direction, which may further improve the generation performance of DPDM with the same sampling steps.

**Training Efficiency and GPU-memory Cost.** Another characteristic of the DPDM is its good training efficiency and moderate GPU memory requirements. Take the CIFAR10 dataset as an instance, we find that the DPDM's FID (7 sampling steps) converges to 2.72 within 100k training iterations with a batch size of 128. The wall-clock time that it costs is about 13 hours on a cluster with 4 Nvidia 2080ti GPUs and with PyTorch distributed data-parallel framework, which is considered not time expensive. As for the training memory costs, we compare the memory costs of DPDM with consistency distillation as a baseline. We summarize the GPU and CPU statistics during training in Table 6. Both models are trained on a 2 Nvidia 2080ti GPU with PyTorch DDP framework. In conclusion, the training of DPDM consumes more GPU memory costs and almost the same CPU memory as CD. This is due to that the training of DPDM involves deploying an auxiliary diffusion model as we introduced in algorithm 1. Fortunately, the auxiliary DM and the DPDM are updated alternatively, therefore, the training does not need to backpropagate through a large computational graph that involves both the auxiliary DM and the DPDM. As a result, the additional training memory cost is moderate and acceptable.

## 6 CONCLUSION AND LIMITATIONS

In this work, we have presented a novel multi-step generative model that we call the data-prediction denoising model (DPDM). We have shown that empirically, DPDM has a stronger denoising ability than diffusion models. We have also shown that empirically DPDM outperforms other multi-step generative models such as DMs, PDs, CMs, and Denoising-Diffusion GAN models under the few-step setting (less than 10 NFEs when sampling). In conclusion, the DPDM has been shown to be a promising multi-step diffusion-based generative model.

Nonetheless, DPDM has its limitations that call for further research along this line. First, despite its promising generative ability, DPDM has shown sub-optimal performance than diffusion models if the sampling steps are set to be sufficient. This limitation motivates us to further improve DPDM in future work. Second, the training of DPDM relies on an auxiliary diffusion model, which may bring additional memory costs. We hope our exploration of DPDM could shed some light and contribute to developing more efficient and powerful multi-step generative models for the community in the future.

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

# A  TECHNICAL DETAILS

## A.1  ROBUSTNESS OF SMOOTHED KL DIVERGENCE

One of the benefits of the smoothed KL divergence over the traditional KL divergence is its robustness to misaligned density support. To illustrate this advantage, we consider a well-known example in Arjovsky et al. (2017). Let $z$ be a random variable following the uniform distribution on the unit interval $[0,1]$. Consider $\mathbb{P}_0$ to be the distribution of $(0,z) \in \mathbb{R}^2$. Now let $g_\theta(z) = (\theta,z)$, where $\theta$ is a single real parameter. The density of $\mathbb{P}_0$ and $\mathbb{P}_\theta$ are $p_0(x,z) = \mathbb{I}_{x=0}(x)\mathbb{I}_{[0,1]}(z)$ and $p_\theta(x,z) = \mathbb{I}_\theta(x)\mathbb{I}_{[0,1]}(z)$. Since for each $\theta \neq 0$, the support of $p_\theta$ and $p_0$ does not intersect, the KL divergence between $\mathbb{P}_\theta$ and $\mathbb{P}_0$ is ill-defined with

$$\mathcal{D}_{KL}(\mathbb{P}_\theta,\mathbb{P}_0) = \begin{cases} +\infty & \theta \neq 0; \\ 0 & \theta = 0. \end{cases} \tag{A.1}$$

The same is also true for the Jensen-Shannon divergence where

$$\mathcal{D}_{JS}(\mathbb{P}_\theta,\mathbb{P}_0) = \begin{cases} \log 2 & \theta \neq 0; \\ 0 & \theta = 0. \end{cases} \tag{A.2}$$

So minimizing the KL divergence with a gradient-based algorithm does not lead the generator to converge to the correct parameter $\theta = 0$. However, the smoothed KL divergence (defined in (3.2)) provides a finite and reliable objective for training the generator. More precisely, considering the simple diffusion defined in (2.3): $x_t = x_0 + \sigma(t)\epsilon, \epsilon \sim \mathcal{N}(0,I)$. The marginal distribution of $(x,z)$ initialized with $\mathbb{P}_0$ and $\mathbb{P}_\theta$ writes

$$p_0^{(t)}(x,z) = \mathcal{N}(x;0,t)\int_0^1 \mathcal{N}(z;s,t)\mathrm{d}s,$$

$$p_\theta^{(t)}(x,z) = \mathcal{N}(x;\theta,t)\int_0^1 \mathcal{N}(z;s,t)\mathrm{d}s,$$

which are defined on $(x,z) \in \mathbb{R}^2$. The notation $\mathcal{N}(x;\mu,\sigma^2)$ represents the density function of Gaussian distribution with mean $\mu$ and variance $\sigma^2$. The smoothed KL divergence with weight function thus has the expression

$$\begin{aligned}
\mathcal{D}_{smooth-KL}(\mathbb{P}_\theta,\mathbb{P}_0) &= \mathbb{E}_{(x,z)\sim p_\theta^{(t)}(x,z)} \log \frac{q_{(t,\theta)}(x,z)}{p_\theta^{(0)}(x,z)} \\
&= \mathbb{E}_{(x,z)\sim p_\theta^{(t)}(x,z)} \log \frac{\mathcal{N}(x;\theta,t)}{\mathcal{N}(x;0,t)} \\
&= \mathbb{E}_{x\sim\mathcal{N}(x;\theta,t)} \log \frac{\mathcal{N}(x;\theta,t)}{\mathcal{N}(x;0,t)} \\
&= \mathbb{E}_{x\sim\mathcal{N}(x;\theta,t)} \frac{1}{2t}\left[x^2 - (x-\theta)^2\right] \\
&= \mathbb{E}_{x\sim\mathcal{N}(x;\theta,t)} \frac{1}{2t}\left[2\theta x - \theta^2\right] \\
&= \frac{1}{2t}\theta^2, \theta \in \mathbb{R}.
\end{aligned} \tag{A.3}$$

For each $t > 0$, $\mathcal{D}_{smoothed-KL}$ is finite. The smoothed KL divergence in (A.3) is a differentiable quadratic function of parameter $\theta$ with a single minima $\theta = 0$ which lead to the $\mathcal{D}_{smooth-KL}(\mathbb{P}_\theta,\mathbb{P}_0) = 0$. Table 5 shows a summary of the comparison among smooth KL divergence, KL divergence, and the Wasserstein distance between $\mathbb{P}_\theta$ and $\mathbb{P}_0$. The smooth KL is more suitable for learning $\theta$ with gradient-based optimization algorithms.

## A.2  THE PROOF OF THEOREM 3.2

**Theorem A.1.** Let $s_{q_t}(x_t) := \nabla_{x_t}\log q_t(x_t)$ and $s_{p_t}(x_t) := \nabla_{x_t}\log p_t(x_t)$, the gradient of the smoothed KL in (3.2) between $q_0$ and $p_0$ is

$$\mathrm{Grad}(\theta) = \mathbb{E}_{\substack{x_t = d_\theta(x_0+\epsilon)+\tilde{\epsilon} \\ x_0 \sim p_0, \epsilon, \tilde{\epsilon} \sim \mathcal{N}(0,I)}} \left[s_{q_t}(x_t) - s_{p_t}(x_t)\right]\frac{\partial x_t}{\partial \theta}.$$

| Divergence | distance | smooth |
|---|---|---|
| **smooth KL (ours)** | $\propto \theta^2$ | ✓ |
| KL | $+\infty$ (A.1) | ✗ |
| Wasserstein | $\propto |\theta|$ (Arjovsky et al., 2017) | ✗ |
| Jensen-Shannon | $\log 2$ (A.2) | ✗ |

Table 5: Comparison of divergence property against smooth KL.

*Proof.* Recall the definition of $p_t$ and $q_t$: $p_t(\boldsymbol{x}_t)$ denote the distribution of $\boldsymbol{x}_t = \boldsymbol{x}_0 + \epsilon, \boldsymbol{x}_0 \sim p_0(\boldsymbol{x}_0), \epsilon \sim \mathcal{N}(\mathbf{0}, \boldsymbol{I})$, and $q_t(\widehat{\boldsymbol{x}}_t)$ is defined as the distribution of $\widehat{\boldsymbol{x}}_t = \boldsymbol{d}_\theta(\boldsymbol{x}_t) + \widetilde{\epsilon}, \boldsymbol{x}_t \sim p_t(\boldsymbol{x}_t), \widetilde{\epsilon} \sim \mathcal{N}(\mathbf{0}, \boldsymbol{I})$. There are two terms that contain the denoiser's parameter $\theta$. The term $\widehat{\boldsymbol{x}}_t$ contains parameter through $\widehat{\boldsymbol{x}}_t = \boldsymbol{d}_\theta(\boldsymbol{x}_0 + \epsilon) + \widetilde{\epsilon}$. The marginal density $q_t$ also contains parameter $\theta$ implicitly since $q_t$ is initialized with $q_0$ which is recovered by the denoiser $\boldsymbol{d}_\theta$. To demonstrate the parameter dependence, we may use $q_{(t,\theta)}$ to represent $q_t$.

The $p_t$ is defined through the pre-trained diffusion models with score functions $\boldsymbol{s}_{p_t}$. The smoothed KL divergence between $q_t$ and $p_t$ is defined with,

$$\mathcal{D}_{smooth-KL}(q_{0,\theta}, p_0) := \mathcal{D}_{KL}(q_{(t,\theta)}, p_t) = \mathbb{E}_{\boldsymbol{x}_t \sim q_{(t,\theta)}}\left[\log \frac{q_{(t,\theta)}(\boldsymbol{x}_t)}{p_t(\boldsymbol{x}_t)}\right] \tag{A.4}$$

Taking the $\theta$ gradient of smooth KL (A.4), we have

$$\frac{\partial}{\partial \theta} \mathcal{D}_{smooth-KL}(q_{(t,\theta)}, p_t)$$

$$= \frac{\partial}{\partial \theta} \mathbb{E}_{\substack{\boldsymbol{x}_t = \boldsymbol{x}_0 + \epsilon, \\ \boldsymbol{x}_0 \sim p_0(\boldsymbol{x}_0), \epsilon, \widetilde{\epsilon} \sim \mathcal{N}(\mathbf{0}, \boldsymbol{I})}} \left[\log \frac{q_{(t,\theta)}(\boldsymbol{d}_\theta(\boldsymbol{x}_t) + \widetilde{\epsilon})}{p_t(\boldsymbol{d}_\theta(\boldsymbol{x}_t) + \widetilde{\epsilon})}\right]$$

$$= \mathbb{E}_{\substack{\boldsymbol{x}_t = \boldsymbol{x}_0 + \epsilon, \\ \boldsymbol{x}_0 \sim p_0(\boldsymbol{x}_0), \epsilon, \widetilde{\epsilon} \sim \mathcal{N}(\mathbf{0}, \boldsymbol{I})}} \frac{\partial}{\partial \theta} \left[\log \frac{q_{(t,\theta)}(\boldsymbol{d}_\theta(\boldsymbol{x}_t) + \widetilde{\epsilon})}{p_t(\boldsymbol{d}_\theta(\boldsymbol{x}_t) + \widetilde{\epsilon}))}\right]$$

$$= \mathbb{E}_{\substack{\boldsymbol{x}_t = \boldsymbol{x}_0 + \epsilon, \\ \boldsymbol{x}_0 \sim p_0(\boldsymbol{x}_0), \epsilon, \widetilde{\epsilon} \sim \mathcal{N}(\mathbf{0}, \boldsymbol{I})}} \nabla_{\boldsymbol{x}_t} \left[\log \frac{q_{(t,\theta)}(\boldsymbol{d}_\theta(\boldsymbol{x}_t) + \widetilde{\epsilon})}{p_t(\boldsymbol{d}_\theta(\boldsymbol{x}_t) + \widetilde{\epsilon}))}\right] \frac{\partial \left[\boldsymbol{d}_\theta(\boldsymbol{x}_t) + \widetilde{\epsilon}\right]}{\partial \theta}$$

$$+ \mathbb{E}_{\substack{\boldsymbol{x}_t = \boldsymbol{x}_0 + \epsilon, \widehat{\boldsymbol{x}}_t = \boldsymbol{d}_\theta(\boldsymbol{x}_t) + \widetilde{\epsilon} \\ \boldsymbol{x}_0 \sim p_0(\boldsymbol{x}_0), \epsilon, \widetilde{\epsilon} \sim \mathcal{N}(\mathbf{0}, \boldsymbol{I})}} \frac{\partial}{\partial \theta} \log q_{(t,\theta)}(\widehat{\boldsymbol{x}}_t)|_{\widehat{\boldsymbol{x}}_t = \boldsymbol{d}_\theta(\boldsymbol{x}_t) + \widetilde{\epsilon}}$$

$$= \mathbb{E}_{\substack{\boldsymbol{x}_t = \boldsymbol{x}_0 + \epsilon, \widehat{\boldsymbol{x}}_t = \boldsymbol{d}_\theta(\boldsymbol{x}_t) + \widetilde{\epsilon} \\ \boldsymbol{x}_0 \sim p_0(\boldsymbol{x}_0), \epsilon, \widetilde{\epsilon} \sim \mathcal{N}(\mathbf{0}, \boldsymbol{I})}} \nabla_{\widehat{\boldsymbol{x}}_t} \left[\log \frac{q_{(t,\theta)}(\widehat{\boldsymbol{x}}_t)}{p_t(\widehat{\boldsymbol{x}}_t))}\right] \frac{\partial \widehat{\boldsymbol{x}}_t}{\partial \theta} + \mathbb{E}_{\widehat{\boldsymbol{x}}_t \sim q_{(t,\theta)}} \frac{\partial}{\partial \theta} \log q_{(t,\theta)}(\widehat{\boldsymbol{x}}_t)$$

$$= A + B. \tag{A.5}$$

The term $A$ in equation (A.5) writes

$$A = \mathbb{E}_{\widehat{\boldsymbol{x}}_t \sim q_{(t,\theta)}} \left[\boldsymbol{s}_{q_t}(\widehat{\boldsymbol{x}}_t) - \boldsymbol{s}_{p_t}(\widehat{\boldsymbol{x}}_t)\right] \frac{\partial \boldsymbol{x}_t}{\partial \theta} \tag{A.6}$$

We show that the term $B$ in equation (A.5) vanishes.

$$B = \mathbb{E}_{\widehat{\boldsymbol{x}}_t \sim q_{(t,\theta)}} \frac{\partial}{\partial \theta} \log q_{(t,\theta)}(\widehat{\boldsymbol{x}}_t)$$

$$= \int \frac{1}{q_{(t,\theta)}(\widehat{\boldsymbol{x}}_t)} \frac{\partial}{\partial \theta} q_{(t,\theta)}(\widehat{\boldsymbol{x}}_t) q_{(t,\theta)}(\widehat{\boldsymbol{x}}_t) \mathrm{d}\widehat{\boldsymbol{x}}_t$$

$$= \int \frac{\partial}{\partial \theta} q_{(t,\theta)}(\widehat{\boldsymbol{x}}_t) \mathrm{d}\widehat{\boldsymbol{x}}_t$$

$$= \frac{\partial}{\partial \theta} \int q_{(t,\theta)}(\widehat{\boldsymbol{x}}_t) \mathrm{d}\widehat{\boldsymbol{x}}_t \tag{A.7}$$

$$= \frac{\partial}{\partial \theta} \mathbf{1} \mathrm{d}\widehat{\boldsymbol{x}}_t$$

$$= \mathbf{0}$$

$$\tag{A.8}$$

The equality (A.7) holds if function $q_{(t,\theta)}(\boldsymbol{x})$ satisfies the conditions (1). $q_{(t,\theta)}(\boldsymbol{x})$ is Lebesgue integrable for $\boldsymbol{x}$ with each $\theta$; (2). For almost all $\boldsymbol{x} \in \mathbb{R}^D$, the partial derivative $\partial q_{(t,\theta)}(\boldsymbol{x})/\partial\theta$ exists for all $\theta \in \Theta$. (3) there exists an integrable function $h(.) : \mathbb{R}^D \to \mathbb{R}$, such that $q_{(t,\theta)}(\boldsymbol{x}) \leq h(\boldsymbol{x})$ for all $\boldsymbol{x}$ in its domain. Then the derivative w.r.t $\theta$ can be exchanged with the integral over $\boldsymbol{x}$, i.e.

$$\int \frac{\partial}{\partial\theta} q_{(t,\theta)}(\boldsymbol{x}) \mathrm{d}\boldsymbol{x} = \frac{\partial}{\partial\theta} \int q_{(t,\theta)}(\boldsymbol{x}) \mathrm{d}\boldsymbol{x}.$$

$\square$

### A.3 MORE DISCUSSIONS ON RELATED WORKS

**Compare DPDM with Other Multi-step Models.** Luo et al. (2023) proposed the Diff-Instruct, a strong data-free method that can distill the teacher diffusion model into one-step student generative models. The foundation of Diff-Instruct is based on minimizing an integral Kullback-Leibler divergence, which shares a similar form of our gradient expression (3.4) of DPDM. However, the DPDM has fundamentally different concepts from Diff-Instruct. First, the Diff-Instruct only supports the training of a one-step implicit generative model, i.e. a direct generator. Instead, the DPDM aims to train a series of strong denoiser models, which are composed to form a multi-step generative model. Second, since the training of DPDM is based on denoising noisy data, it requires the ground truth dataset. On the contrary, the Diff-Instruct does not rely on the ground truth dataset for training one-step generative models. Third, as we show in Table 3 in Section 5.2, the generative performance of multi-step DPDM is significantly better than the one-step generator trained with Diff-Instruct on each benchmark. This shows the benefits of multi-step generative modeling over the single-step model. Fourth, the trained denoisers of DPDM are able to be used for tasks, such as data distribution recovery, as we introduced in Section 5.1, for which the Diff-Instructed one-step generator can not be used.

The progressive Distillation (PD (Salimans and Ho, 2022)) and Denoising Diffusion GANs (DD-GAN) (Xiao et al., 2021) are two other representative multi-step generative models. These two models use neural mappings to implement the distribution transfers between intermediate consecutive marginal distributions of the diffusion processes. Therefore, their inference does not support the flexible choice of sampling steps. This means that if a PD or DD-GAN model is trained to support 8 steps, it can not support the inference with more or less than 8 steps. The DPDM follows a different concept from PD and DD-GAN. The DPDM directly learns the mapping that transfers noisy data to clean data. This makes the DPDM highly flexible because the model's multiple inference steps all lead to the data (instead of inter-mediate noisy data). Therefore, DPDM can support arbitrary sampling steps in practical uses.

**A Detailed Comparison with Diff-Instruct** Luo et al. (2023) proposed the Diff-Instruct, a strong data-free method that can distill the teacher diffusion model into one-step student generative models. Let $\boldsymbol{s}_{p_t}$ denote the marginal score functions provided by pre-trained teacher diffusion models, where $p_t(\boldsymbol{x}_t)$'s are the underlying diffused data distributions at time $t$ according to (2.1). Assume $\boldsymbol{s}_{p_t}$ provides a sufficiently good approximation of diffused data distribution. The goal of Diff-Instruct is to use score functions provided by teacher DM to train a one-step student, i.e. a generator that takes a random noise $\boldsymbol{z} \sim p_z$ to obtain generated data $\boldsymbol{x}_0 = g_\theta(\boldsymbol{z})$, where $g_\theta$ is a neural network with the same output dimension as the data. Let $q_t = q_g$ denote the distribution of the one-step student generator, and $\boldsymbol{s}_{q_t}(\boldsymbol{x}_t) := \nabla_{\boldsymbol{x}_t} \log q_t(\boldsymbol{x}_t)$ be the marginal score functions of the generator distribution after forward diffusion. Luo et al. (2023) shows that the one-step student can be trained by minimizing the Integral Kullback-Liebler (IKL) divergence between the generator and the teacher diffusion model's distribution as defined in (A.9)

$$\mathcal{D}_{IKL}^{[0,T]}(q_g, p_d) := \int_{t=0}^{T} w(t) \mathcal{D}_{KL}(q_t, p_t) \mathrm{d}t := \int_{t=0}^{T} w(t) \mathbb{E}_{\boldsymbol{x}_t \sim q_t} \Big[ \log \frac{q_t(\boldsymbol{x}_t)}{p_t(\boldsymbol{x}_t)} \Big] \mathrm{d}t, \tag{A.9}$$

They show that the parameter gradient to minimize the IKL has an explicit formula (A.10)

$$\mathrm{Grad}(\theta) = \int_{t=0}^{T} w(t) \mathbb{E}_{\substack{\boldsymbol{z} \sim p_z, \boldsymbol{x}_0 = g_\theta(\boldsymbol{z}), \\ \boldsymbol{x}_t | \boldsymbol{x}_0 \sim p_t(\boldsymbol{x}_t | \boldsymbol{x}_0)}} \big[ \boldsymbol{s}_{q_t}(\boldsymbol{x}_t) - \boldsymbol{s}_{p_t}(\boldsymbol{x}_t) \big] \frac{\partial \boldsymbol{x}_t}{\partial\theta} \mathrm{d}t. \tag{A.10}$$

In practice, Luo et al. (2023) shows that the generator score function $\boldsymbol{s}_{q_t}(\boldsymbol{x}_t)$ is estimated via fine-tuning an auxiliary diffusion model $\boldsymbol{s}_\phi(\boldsymbol{x}_t, t)$ as proposed in Luo et al. (2023). The Diff-Instruct achieves strong performance when distilling diffusion models to a single-step generator, leading to

extreme acceleration of DMs. However, it has its own limitation: it can only distill diffusion models to one-step generators, but for complex data distributions, a single-step generator is not powerful enough to maintain the strong performance of the diffusion models. A preferred generative model is usually assumed to support both single-step and multi-step generations to trade-off efficiency and strong performance.

The Diff-Instruct has shown promising results, which are capable of training a one-step student model that is comparable to multi-step teacher diffusion models and, therefore achieves tens of sampling accelerations. However, the Diff-Instruct can only distill the teacher diffusion model into one-step generators. Therefore the performance and flexibility are limited by the one-step student's architecture. Though the goal of Diff-Instruct and DPDM is very different, in our work, we borrow some common concepts from Diff-Instruct when introducing the training algorithm of our DPDM.

Furthermore, DPDM offers very high flexibility to the generator, distinguishing it from traditional diffusion distillation methods that impose strict constraints on the generator selection. For instance, the generator can be a convolutional neural network (CNN)-based or a Transformer-based image generator such as StyleGAN (Karras et al., 2019; 2020; 2021; Lee et al., 2021), or an UNet-based generator (Xiao et al., 2021) adapted from pre-trained diffusion models (Karras et al., 2022; Song et al., 2020b; Ho et al., 2020). The versatility of DPDM allows it to be adapted to different types of generators, expanding its applicability across a wide range of generative modeling tasks. In the experiment sections, we show that DPDM is capable of transferring knowledge to generator architectures including both UNet-based and GAN generators respectively. To the best of our knowledge, the DPDM is the first approach to efficiently enable such a data-free knowledge transfer from diffusion models to generic implicit generators.

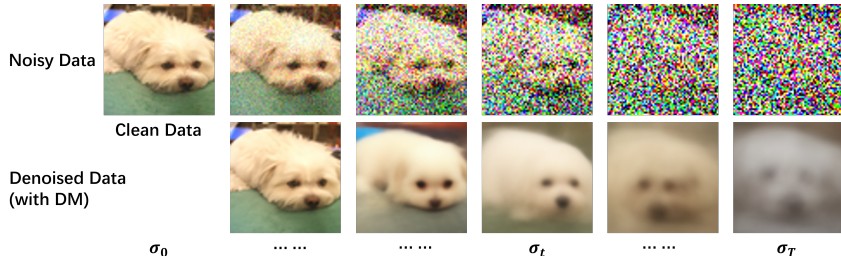

Figure 2: Demonstration of the denoising effect of diffusion model's denoisers. When the variance of the added Gaussian noise grows, the DM's denoisers quickly become unable to recover the clean data (and the data distribution). This drawback limits the DM's generation performance when sampling steps are limited to be few. **Upper:** noisy data; **Under:** denoised data by diffusion model's corresponding denoisers.

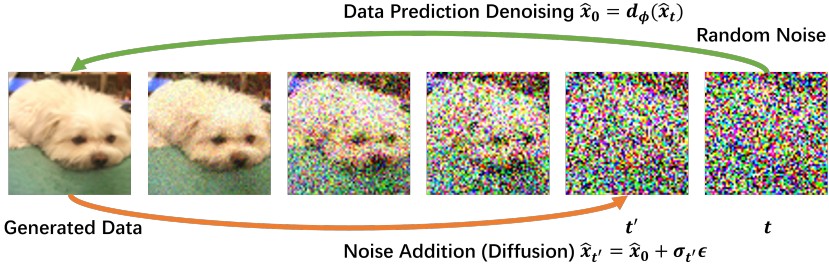

Figure 3: Demonstration of the DMDP's sampling algorithm 2. The algorithm consists of successive iterations of a data prediction denoising step and a Gaussian noise addition step.

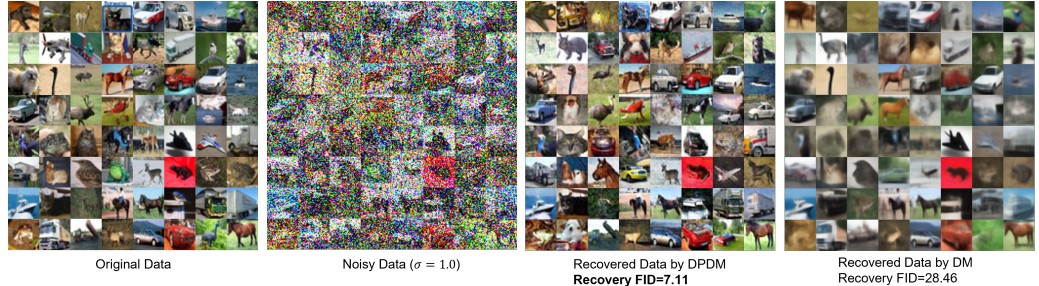

Figure 4: Visualization of Data Recovery performance of DPDM and DM.

Table 6: Statistics of training memory costs of DPDM on Cifar10 datasets.

| Model/ Stats | Peak GPU-Memo(GB) | Peak CPU-Memo(GB) | Sec-per-K Iterations |
|---|---|---|---|
| CD | 9.55 | 2.75 | 0.0489 |
| DPDM | 10.40 | 2.78 | 0.0728 |

## A.4 ALGORITHMS, TABLES AND FIGURES

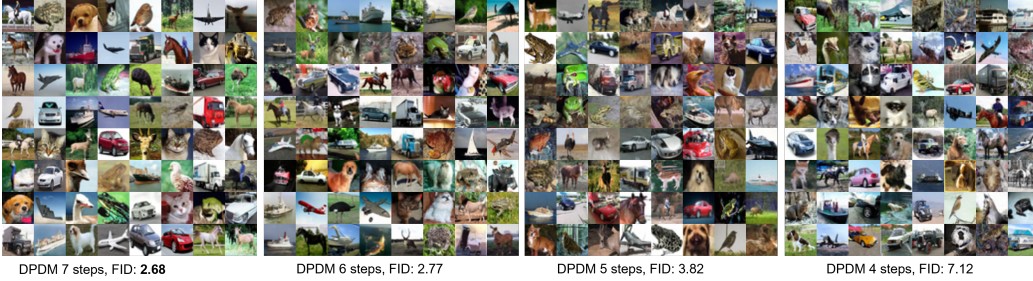

DPDM 7 steps, FID: **2.68**    DPDM 6 steps, FID: 2.77    DPDM 5 steps, FID: 3.82    DPDM 4 steps, FID: 7.12

Figure 5: Comparison of performance of DPDM with different sampling steps on CIFAR10 unconditional generation.

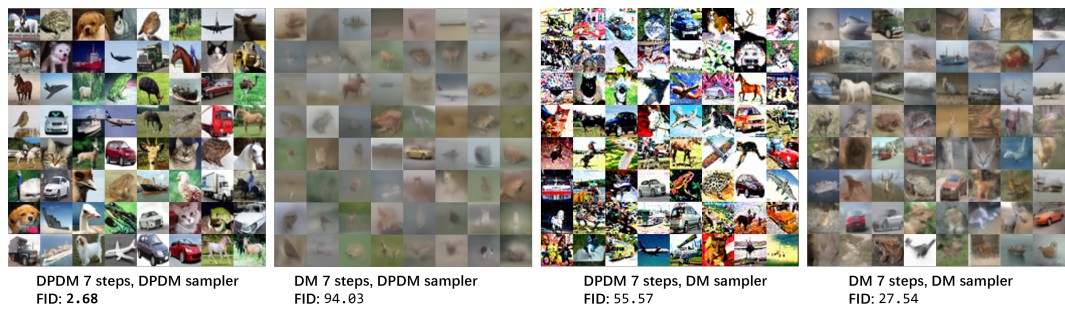

DPDM 7 steps, DPDM sampler    DM 7 steps, DPDM sampler    DPDM 7 steps, DM sampler    DM 7 steps, DM sampler
FID: **2.68**                 FID: 94.03                   FID: 55.57                  FID: 27.54

Figure 6: Comparison of samples from DPDM and DM with different sampling algorithms on CIFAR10 unconditional generation.

## B  MORE ON EXPERIMENTS

**Settings.**    To demonstrate the efficacy of our proposed DPDM, We choose the state-of-the-art EDMs Karras et al. (2022) as baselines, which have achieved state-of-the-art generative performance on several benchmarks such as CIFAR10 and ImageNet $64 \times 64$. The EDM model depends on the diffusion process

$$d\boldsymbol{x}_t = G(t)d\boldsymbol{w}_t, t \in [0,T]. \tag{B.1}$$

Samples from the forward process (B.1) can be generated by adding random noise to the output of the generator function, i.e., $\boldsymbol{x}_t = \boldsymbol{x}_0 + \sigma(t)\boldsymbol{\epsilon}$ where $\boldsymbol{\epsilon} \sim \mathcal{N}(\boldsymbol{0}, \boldsymbol{I})$ is a Gaussian vector and $\sigma(t) := \sqrt{\int_0^t G^2(s)ds}$ is a function with explicit expressions. We download the pre-trained model checkpoints from the official website[1] and consider transferring their knowledge to implicit generative models, specifically UNet and StyleGAN as generators.

We compare the DPDM with the baseline EDM (Karras et al., 2022) models and other diffusion-based multi-step generative models. We evaluate the performance of the trained DPDM via FID. We calculate the FID in the same way as the StyleGAN2-ADA[2] codebase. For the ImageNet $64 \times 64$ dataset, we use the same pre-processing scripts as the EDM model on the ImageNet $64 \times 64$ dataset.

### B.1  DETAILED EXPERIMENTAL SETTINGS OF FEW-STEP IMAGE GENERATION

**Sampling Configurations.**    The sampling algorithm 2 relies on a pre-defined sequence of standard deviations. In our implementation, we follow a similar concept of EDM sampler (Karras et al., 2022), in which the discretized noise levels in algorithm 2 are chosen by

$$\sigma_{k<K} = \left(\sigma_{max}^{\frac{1}{\rho}} \frac{i}{K-1} (\sigma_{min}^{\frac{1}{\rho}} - \sigma_{max}^{\frac{1}{\rho}})\right)^\rho, \ and \ \sigma_K = 0$$

Here we set $\sigma_{max} = 80.0, \rho = 40.0$ and $\sigma_{min}$ to vary for different sampling steps. The values for $\sigma_{min}$ are put in Table B.1.

---

[1] https://github.com/NVlabs/edm
[2] https://github.com/NVlabs/stylegan2-ada-pytorch

Table 7: Choice of hyper-parameters $\sigma_{min}$ of sampling algorithm for different NFEs.

| $\sigma_{min}/\downarrow$/NFE | 4 | 5 | 6 | 7 | 8 | 9 | 10 | 12 | 14 | 16 | 18 |
|---|---|---|---|---|---|---|---|---|---|---|---|
| $\sigma_{min}$ | 0.3 | 0.2 | 0.1 | 0.015 | 0.0035 | 0.00055 | 0.0001 | 0.0001 | 0.0001 | 0.0001 | 0.0001 |

Table 8: Hyperparameters used for training DPDM

| Hyperparameter | CIFAR-10 (Uncond) | | ImageNet $64 \times 64$ | | CIFAR-10 (Cond) | |
|---|---|---|---|---|---|---|
| | DM $s_\phi$ | DPDM $d_\theta$ | DM $s_\phi$ | DPDM $d_\theta$ | DM $s_\phi$ | DPDM $d_\theta$ |
| Learning rate | 1e-5 | 1e-5 | 1e-5 | 1e-5 | 1e-5 | 1e-5 |
| Batch size | 128 | 128 | 96 | 96 | 128 | 128 |
| $Adam\ \beta_0$ | 0.0 | 0.0 | 0.0 | 0.0 | 0.0 | 0.0 |
| $Adam\ \beta_1$ | 0.99 | 0.99 | 0.99 | 0.99 | 0.99 | 0.99 |
| Training iterations | 100k | 100k | 50k | 50k | 100k | 100k |
| Number of GPUs | 4 | 4 | 8 | 8 | 4 | 4 |

**Detailed Performances.** Tables 2 and 3 summarize the FID values of DPDM, the baseline EDMs, along with other multi-step generative models on the CIFAR10 datasets (unconditional without labels) and the conditional generation on the ImageNet $64 \times 64$ data. The notation NFE in both tables refers to the number of neural function evaluations, which represents the number of sampling steps for a multi-step model. On both datasets, DPDM performs significantly better than both baseline diffusion models and other diffusion-based multi-step generative models under the few NFE settings (NFE no larger than 10). Besides, DPDMs perform competitively across all datasets among multiple-step generative models, which involve both models from diffusion distillation or direct training.

As shown in Table 3 and Table 2, on the ImageNet dataset of the resolution of $64 \times 64$, DPDM outperforms diffusion-based single-step generative models in terms of FID, achieving a state-of-the-art few-step generation performance among diffusion-based models with an FID of 3.61 and an NFE of 10. On the unconditional generation of the CIFAR10 dataset, DPDM achieves the state-of-the-art few-step FID value among diffusion-based generative models, with an FID of 2.68 and an NFE of 8. It outperforms other mult-step generative models such as consistency distillation (CD) and the consistency training (Song et al., 2023), the progressive distillation (Salimans and Ho, 2022), the Diff-Instruct (Luo et al., 2023), the DD-GAN (Xiao et al., 2021) and the diffusion model with various sampling algorithms such as SA-SolverXue et al. (2023), UniPC(Zhao et al., 2023), and the DPM-Solver (Lu et al., 2022). It is worth noting that the CD, CT, Diff-Instruct, PD, and EDM with SDE/ODE samplers all share the same model architecture and the teacher diffusion model (if necessary), therefore the comparison with these models especially highlights the promising few-step generation ability of DPDM.

According to Table 2, the 5-step generation performance of DPDM (with an FID of 3.85) is better than the 15-step generation performance of the baseline EDM model (with an FID of 5.62). This indicates that DPDM achieves about 3 times more efficiency than baseline diffusion models with competitive performances. Figure 1 shows some non-cherry-picked few-step generated samples from DPDM and baseline EDM models on the ImageNet$64 \times 64$ and the CIFAR10 datasets. The DPDM data is generated with sampling algorithms 2, while the DM samples are generated with DM's default ODE samplers. We find that using DPDM's sampling algorithm for DM leads to worse performance than DM's original sampling algorithms. As is shown in Figure 1, the DPDM is able to generate high-quality samples with good diversities with no more than 10 NFEs. On the contrary, the diffusion model fails to generate satisfactory samples with such few sampling steps. In conclusion, the DPDM achieves promising few-step generative performance under challenging few-step settings.

**Other training details.** Following Luo et al. (2023), we initialize the auxiliary diffusion model $s_\phi(.,.)$ with the weight parameters of pre-trained DM. We train the DPDM and auxiliary model on a 4-GPU cluster with Nvidia 2080ti GPUs. We implemented the training program with PyTorch of version 1.12.1 and the default distributed data-parallel mechanism for parallel training. During training, we use our DPDM algorithm 1 to update the auxiliary DM and the DPDM's parameters. We use the Adam optimizer for both the DM and the generator. For the DPDM, we use the same exponential moving average (EMA) technique as the EDM model's training scripts. Table 8 summarizes some hyper-parameters for training DPDMs on CIFAR10 and ImageNet$64 \times 64$ datasets.

**Training Efficiency and GPU-memory Cost.** Another characteristic of the DPDM is its good training efficiency and moderate GPU memory requirements. More precisely, on CIFAR10 datasets, we find that the DPDM's generation FID (7 sampling steps) converges to 2.72 within 100k training iterations with a batch size of 128. The wall-clock time that the training costs is about 13 hours on a cluster with 4 Nvidia 2080ti GPUs and with PyTorch distribution data parallel training framework. Overall, we find empirically that the training time costs are comparable with consistency distillation. As for the training memory costs, we compare the memory costs of DPDM with baseline consistency distillation. We summarize the GPU and CPU statistics during training in Table 6. Both models are trained on a 2 Nvidia 2080ti GPU with PyTorch DDP parallel framework. In conclusion, the training of DPDM consumes more GPU memory costs and almost the same CPU memory as CD. This is due to that the training of DPDM involves deploying an auxiliary diffusion model as we introduced in algorithm 1. Fortunately, the auxiliary DM and the DPDM are updated alternatively, therefore, the training does not need to backpropagate through a large computational graph that involves both the auxiliary DM and the DPDM. As a result, the additional training memory cost is moderate and acceptable.

