# OpenReview forum: "Data Prediction Denoising Models: The Pupil Outdoes the Master"
_ICLR.cc/2024/Conference — Submitted to ICLR 2024_

### Official Review · Reviewer_gGZJ · 2023-10-30

**Soundness:** 3 good
**Presentation:** 3 good
**Contribution:** 3 good
**Rating:** 8
**Confidence:** 2

**Summary:**

This paper addresses the performance drop in DMs with limited sampling steps, attributing it to the weak denoisers used in their training. To mitigate this issue, the authors introduce the Data Prediction Denoising Model (DPDM), a multi-step generative model that outperforms DMs with few sampling steps. DPDM enhances data recovery capabilities by minimizing distribution divergence, which results in stronger denoisers capable of better recovering data distributions from noisy data. A corresponding sampling algorithm, DPDM sampler, is introduced to generate samples from DPDMs.

**Strengths:**

1) The paper addresses a useful and practical issue with Diffusion Models (DMs). The research is novel and the exposition is clear and well-structured.
2) The paper begins with a clear and well-supported empirical observation regarding the performance drop in DMs when the number of sampling steps is limited. The paper introduces the Data Prediction Denoising Model (DPDM) as a novel approach to address the limitations of DMs. DPDM is designed to enhance data recovery abilities, and its effectiveness is rigorously demonstrated through experiments.
3) The paper provides some mathematical foundation for DPDM by emphasizing the importance of minimizing distribution divergence. This adds depth and theoretical support to the proposed model.
4) The paper goes beyond presenting DPDM and conducts extensive comparisons with existing multi-step generative models.

**Weaknesses:**

1) While the paper claims to solve a practical problem by needing less compute resources for sampling compared to DMs. However, training DPDMs require an auxiliary diffusion model, a retrained DM and a multi-step denoiser model which may bring additional memory costs and training time requirements.
2) The paper predominantly focuses on low-resolution image generation tasks, such as 64x64 pixel images. While it demonstrates the effectiveness of DPDM in this context, it does not explore or provide results for higher-resolution image generation or other types of data generation tasks.
3) Minor comment: When NFE is first introduced in Page 2, the full form is not mentioned. Please include this in the revised version to improve readability.

**Questions:**

While the paper is generally well-written, I have the following questions.

1) Can the authors demonstrate the results for generating higher resolution images such as 256x256 or higher and how DPDMs compare with DMs?
2) Can the authors comment on the training and inference times and compute requirements for DPDMs and comparable DMs?

---

> ### Author Response · Authors · 2023-11-14
> **Thank you for liking our work!**
>
> Thank you for liking our work, we will incorporate your wonderful suggestions in the revision. We will address your concerns one by one in the following paragraphs.
>
> **Q1**. training DPDMs requires an auxiliary diffusion model, a retrained DM, and a multi-step denoiser model which may bring additional memory costs and training time requirements.
>
> **A1**. We appreciate your keen intuition. We acknowledge that the additional diffusion model ($\boldsymbol{s}_ \phi$ in Algorithm 1) brings additional memory cost. However, this additional memory cost is limited because the memory bottleneck of training lies in the computational graph of the backpropagation, instead of only saving one more model. In DPDM's training, the auxiliary model $\boldsymbol{s}_ \phi$ and the model $\boldsymbol{d}_ \theta$ are updated alternatively, which means that the other model's parameters are fixed and do not participate in back-propagation when one model is being updated. So the memory cost for back-propagating through the computational graph is almost the same as one model. To quantitatively measure how much additional computational costs are brought in, we compare DPDM's training with another multi-step model, the consistency model (consistency distillation, CD) in terms of memory and computational costs in Table 6 in the Appendix of the submission.
>
> **Table 6** shows that DPDM brings in minor additional memory costs than CD (10.40 over 9.55). This is because the DPDM only needs additional GPU memory to save the auxiliary model. But $\boldsymbol{s}_\phi$ and $\boldsymbol{d}_ \theta$ are updated alternatively, so their computational graph does not interact. As a result, the memory bottleneck caused by computational graphs and back-propagation does not bring more costs to DPDM.
>
> As for the wall-clock time for 1K iterations, we see that DPDM costs 0.0728 seconds, while the CD costs 0.0489 seconds. This is because each iteration of DPDM consists of two alternate steps as we show in Algorithm 1. Overall, the DPDM costs almost the same GPU and CPU memory as the baseline CD, but about 1.5 times wall-clock time than the CD for each iteration.
>
> **Q2**. it does not explore or provide results for higher-resolution image generation or other types of data generation tasks. Can the authors demonstrate the results for generating higher resolution images such as 256x256 or higher and how DPDMs compare with DMs?
>
> **A2**. We appreciate your suggestions on larger-scale experiments such as high-res image modeling or text2img generation. However, these experiments demand substantial computational resources beyond our current reach. For example, training a Stable Diffusion t2i model consumes a cloud cluster with 256 Nvidia A100 GPUs for about 150,000 hours, costing a total of about 600,000 USD, which is far beyond our financial affordance. However, in this work, we have demonstrated strong performances of DPDM on standard benchmarks, and we are glad to try to explore the possibility of scaling DPDM to larger-scale tasks such as high-res data generation and t2i generative models within our financial permissions.
>
> **Q3**. Can the authors comment on the training and inference times and compute requirements for DPDMs and comparable DMs?
>
> **A3**. The training of DPDM starts with an initialization of a pre-trained DM. We take the CIFAR10 dataset as a demonstration. Empirically, a DM on the CIFAR10 dataset is supposed to converge on 4 Nvidia-V100 GPUs within 2 wall-clock days. Starting from a pre-trained DM, we empirically find that a DPDM converges within 12 hours. Therefore, if we already have a DM, the total training time of DPDM is about 1/8 of a DM. If we do not have a pre-trained DM, therefore, we are supposed to first pre-train a DM and then train a DPDM. The total time for training DPDM from scratch costs about 9/8 of diffusion models.
>
> As for the inference. We implement the DPDM with the same architecture as the diffusion model (i.e. a weight-sharing network $\boldsymbol{d}_\theta(x_t,t)$ to parametrize multiple denoisers), therefore, the inference cost per step of DPDM is the same as the diffusion model, so NFE is a fair metric for evaluating the inference efficiency when comparing DPDM and the DM. **Table 4** in the submission has a relatively clear comparison of the inference efficiency of DPDM and DM. In Table 4, the FID value of DPDM converges below 3.0 (which is regarded as a line of good performance) with 6 NFEs, on the contrary, the baseline diffusion model's FID is still larger than 3.0 with even 18 NFEs. Since both models have the same cost for each NFE, we conclude that the DPDM is about 3 times more efficient than diffusion models for few-step generations.
>
> We really thank you for your useful suggestions on notations and clarifications which we will incorporate in the revision. We hope our answers can resolve your concerns.

---

### Official Review · Reviewer_twcE · 2023-10-31

**Soundness:** 3 good
**Presentation:** 3 good
**Contribution:** 2 fair
**Rating:** 5
**Confidence:** 3

**Summary:**

This paper introduces an algorithm to train a denoiser given a pre-trained diffusion model, use the denoiser to sample an image, and shows that this method is better than general diffusion on small sampling steps.

**Strengths:**

- A new algorithm to accelerate the diffusion sampling.
- Excellent results on small sampling step.

**Weaknesses:**

With my carefully proofreading, I still suffer from understanding the key idea of this paper. From what I understand, this paper aims to train a separate denoiser such that (1) the denoiser can predict a clean image from noisy image, and (2) when Gaussian noise is added to the predicted clean image, the distribution of the new noisy image is consistent to that of the noisy images used to train the diffusion model. However, training this separate denoiser is the same with the training loss of the original diffusion model. From this aspect, it seems like this idea is more like a fine-tuning (or just training it longer) method. I cannot follow why such a fine-tuning idea is effective when having small sampling step. Could the author clarify this point?

The sampling method considers the clean image as the mean of the next step distribution, which contradicts the theoretical analysis of DDPM. Could the author also justify this point?

Moreover, this paper claims it provides a solid mathematical foundation for the proposed method, which is unclear what that means.

**Questions:**

Need justification and more explanation of the proposed method. (see the weakness)

---

> ### Author Response · Authors · 2023-11-14
> **DPDM is totally different from diffusion models.**
>
> Thank you for your reviews. We will address your concerns one by one in the following paragraphs. Before that, we first give a summary of the main contributions of our work.
>
> In this work, we propose a novel stand-alone generative model termed the Data-prediction denoising model (DPDM), which achieves significantly better generative performance than baseline diffusion models under the few-step sampling settings. We characterize the training process of DPDM through a novel smoothed KL divergence minimization formulation which results in significantly stronger data-recovery ability than the baseline diffusion model.
>
> Next, we address your concerns one by one.
>
> **Q1**. However, training this separate denoiser is the same as the training loss of the original diffusion model. From this aspect, it seems like this idea is more like a fine-tuning (or just training it longer) method. I cannot follow why such a fine-tuning idea is effective when having a small sampling step. Could the author clarify this point?
>
> **A1**. First, we would like to emphasize that our baseline EDM diffusion model is obtained from official checkpoints which are trained till convergence. Second, we would like to emphasize that the training objective of DPDM is totally different from the diffusion model's training. Therefore, the training of DPDM has no relation to fine-tuning with diffusion model loss function.
>
> More precisely, the training objective of the diffusion model is Equation 3.1, which encounters an L2 minimization to train weak denoisers. On the contrary, the DPDM is based on minimizing a novel smoothed KL divergence between the denoiser distribution and the ground truth data distribution. To minimize the smoothed KL for multi-level denoisers, we derive an explicit gradient formula of the denoisers' parameter $\theta$ with respect to a combination of the smoothed KL divergences, i.e. Equation 3.4. Overall, we propose a practical algorithm 1, which implements such a divergence minimization formulation to obtain stronger denoisers.
>
> **Q2**. The sampling method considers the clean image as the mean of the next step distribution, which contradicts the theoretical analysis of DDPM. Could the author also justify this point?
>
> **A2**. We do not clearly understand what you mean by contradicting the theoretical analysis of DDPM. As far as we guess, we assume you have mistaken our sampling algorithm 2 as the same as DDPM's sampling algorithm. We would like to emphasize that the DPDM is a stand-alone generative model that is totally different from diffusion models (so is the DDPM). And its model training and sampling are also different from the diffusion model (DDPM). More precisely, each sampling step of DDPM gradually predicts the mean of the next-step denoised distribution. On the contrary, each step of DPDM's sampling step directly predicts clean data and then adds a Gaussian noise to obtain samples from the less-noisy distribution. Provided that the DPDM is a different generative model, we do not see the necessity of its sampling algorithm to match the DPDM's sampling algorithm.
>
> **Q3**. This paper claims it provides a solid mathematical foundation for the proposed method, which is unclear means?
>
> **A3**. In section 3.2, we elaborately establish the mathematical formulation of enhancing the data-recovery ability of denoisers as a smoothed KL divergence minimization problem. This part highlights the math foundation of DPDM (through probability divergence minimization) and helps to derive a practical training algorithm of DPDM (Algorithm 1).
>
> We hope our answers have resolved your concerns, and if you still have any concerns, please let us know, and we will be glad to further address them.

---

> ### Author Response · Authors · 2023-11-20
> **Thank you for your reviews! we are glad to provide more clarifications.**
>
> Dear reviewer:
>
> Thank you for your reviews! The author-reviewer rebuttal period is coming to a close soon. We sincerely hope that our responses have adequately addressed the concerns you have raised.
>
> In this work, we introduce a **new class of multi-step generative models**, named Data-prediction Denoising Models, which show significantly **stronger performance than diffusion models** under the few-step setting. We also explore the weak data distribution recovery ability of the diffusion model's denoisers and practical solutions for enhancing them. We believe that our findings and solutions may benefit future researchers in either improving diffusion models or proposing new generative models.
>
> We hope our rebuttal has resolved your questions. If you still have any unresolved concerns or additional questions, please do let us know! We would be very glad to provide more clarification and address any remaining issues.
>
> Best wishes,
>
> Authors of the submission #5134.

---

> > ### Comment · Reviewer_twcE · 2023-11-23
> > **Thanks for the rebuttal.**
> >
> > I think the authors's responses have addressed my concerns. I will increase my score accordingly. Two remaining points: (1) please do a better job to **highlight** that DPDM is different from DDPM in the introduction, given DPDM looks quite similar to DDPM (I always worry that I type this method wrong); and (2) like other reviewers, I also have concerns about the application of this method to large-scale problems.

---

> > > ### Author Response · Authors · 2023-11-23
> > > **We are glad that we have addressed your concerns!**
> > >
> > > Thank you for your response!
> > >
> > > We appreciate your keen instinct that our DPDM is easy to be mistaken for the mile-stone work DDPM. We feel sorry for the confusion caused by our DPDM name, which is pronounced quite similar to DDPM. We highly value your suggestions for putting a special paragraph to clearly point out the differences between our DPDM and the DDPM. We will incorporate this constructive suggestion in our revision with a paragraph:
> > >
> > > > The proposed DPDM is essentially different from diffusion models in both training methods and sampling methods.
> > >
> > > > * The training method of diffusion models encounters an image reconstruction objective (a $L^2$ loss function) to train (potentially weak) denoisers. On the contrary, the DPDM is based on minimizing a novel smoothed KL divergence between the denoiser distribution and the ground truth data distribution.
> > >
> > > > * Each sampling step of DDPM gradually predicts the mean of the next-step denoised distribution. On the contrary, each step of DPDM's sampling step directly predicts clean data and then adds a Gaussian noise to obtain samples from the less-noisy distribution.
> > >
> > > > In conclusion, the DPDM is a stand-alone multi-step generative model that differs from diffusion models in both concepts and implementations.
> > >
> > > As for the large-scale problems,  we agree that large-scale experiments will make the submission a very strong acceptance. However, we would like to say that,
> > >
> > > * First, we would like to say, that the main goal of this work is to explore practical methods for developing new multi-step generative models, therefore, we choose to compare the DPDM with other models on CIFAR10 and ImageNet64 which requires minimal additional implementation techniques which may influence the fairness of the comparison of each model (other than tricks) and multi-step samplers.
> > >
> > > * Second, the CIFAR10 and ImageNet64 datasets are the most explored image generation benchmarks, on which nearly all other methods have shown their performances. Therefore, it is suitable and sufficient to have a comprehensive understanding of DPDM on them. Other works, such as EDM, Diff-Instruct, and One-step DEQ, etc., have all conducted comparisons on CIFAR10 and ImageNet64 datasets for comparisons.
> > >
> > > * Third, we think the scaling-up work of DPDM to larger scales, such as the popular text-to-image generation and image-to-image translation applications requires a significant amount of computational resources, engineering implementation, hyper-parameter tuning, and tricks. Frankly speaking, such requirements are often related more to the engineering implementation techniques rather than the generative modeling methodology itself. Therefore, we believe such additional engineering implementations deserve new work in the future.
> > >
> > > However, we highly value your constructive suggestions and we are elaborately trying large-scale dataset experiments, but it may take some time because we do not have sufficient GPU devices;
> > >
> > > In conclusion, we thank you for your constructive suggestions, which are helpful for our work on both the presentation and the impact.
> > >
> > > Best regards,
> > >
> > > Authors of the submission #5134.

---

### Official Review · Reviewer_X9iJ · 2023-11-07

**Soundness:** 3 good
**Presentation:** 3 good
**Contribution:** 3 good
**Rating:** 6
**Confidence:** 3

**Summary:**

This paper proposes Data Prediction Denoising Models (DPDM)s, a new class of generative models that incorporates a sequence of strong denoisers for data generation. It is argued that conventional diffusion models embody weak denoisers, which in turn requires a high number of steps at inference time for generation, reducing the overall efficiency. To alleviate this challenge, DPDM training uses a teacher DM for initialization, and minimizes probability divergences between denoiser-recovered data distributions and the ground truth data distribution. In addition, a sampler suitable for DPDMs is presented. It is shown that DPDM can attain strong performance on CIFAR-10, and ImageNet64x64 with only a few number of sampling steps.

**Strengths:**

- The proposed method is novel, and the technical contribution is strong. DPDMs provide a new framework for generative modeling which is different from diffusion models, although it draws many inspirations from diffusion models.
- The experimental results are convincing where DPDMs outperform many recent baselines, illustrating their capability on data generation using a small number of NFEs.

**Weaknesses:**

- The scope of the experiments section is limited where the results are presented on low-resolution datasets such as CIFAR-10, and ImageNet 64x64. The experiments would benefit from demonstrations with high-resolution datasets to illustrate the generalizability of the method.
- Although the focus of the paper is on inference efficiency, inference metrics are not provided. The comparison is made in terms of NFEs which is an important aspect. However, metrics such as inference memory, seconds per iteration at inference, and overall inference time until convergence should be provided for a valid comparison.

**Questions:**

Questions:
- How does inference memory and time compare with state-of-the-art baselines?
- What does $\tilde{x}$ stand for in Equation 1? I don't think it has been defined anywhere.

Suggestions:
- Typographical issues: There are issues such as duplicate references, (Song et al., 2020b and Song et al., 2020c), duplicate paragraphs (Training Efficiency and GPU-memory Cost in main and appendix) and other issues which should be fixed.

---

> ### Author Response · Authors · 2023-11-14
> **Thank you for your useful suggestions!**
>
> Thank you for your helpful suggestions, we will incorporate them in the revision. We will address your concerns one by one in the following paragraphs.
>
> In this paper, we propose a stand-alone generative model, the DPDM, which draws inspiration from diffusion models from a denoiser perspective. The DPDM shows strong performance under few-step generative settings.
>
> **Q1**. The experiments would benefit from demonstrations with high-resolution datasets to illustrate the generalizability of the method.
>
> **A1**. We appreciate your suggestions on larger-scale experiments such as high-res image modeling or text2img generation. However, these experiments demand substantial computational resources beyond our current reach. For example, training a Stable Diffusion t2i model consumes a cloud cluster with 256 Nvidia A100 GPUs for about 150,000 hours, costing a total of about 600,000 USD, which is far beyond our financial affordance. However, in this work, we have demonstrated strong performances of DPDM on standard benchmarks, and we are glad to try to explore the possibility of scaling DPDM to larger-scale tasks such as high-res data generation and t2i generative models within our financial permissions.
>
> **Q2**. Metrics such as inference memory, seconds per iteration at inference, and overall inference time until convergence should be provided for a valid comparison. How do inference memory and time compare with state-of-the-art baselines?
>
> **A2**. We appreciate your suggestions. First, we would like to say that when implementing DPDM, we use a weight-sharing multi-level denoiser network $\mathbf{d}_\theta(x_t,t)$ to parametrize multiple denoisers in a single network. This network follows the same architecture as the baseline EDM diffusion model. Therefore, the inference costs of DPDM per step are exactly the same as the EDM diffusion model. From this view, the NFE is a fair metric for comparing the sampling efficiency of the DPDM and EDM diffusion models. As for the inference cost for convergence, **Table 4** in the submission has a relatively clear demonstration. In Table 4, the FID value of DPDM converges below 3.0 (which is regarded as a line of good performance) with 6 NFEs, on the contrary, the baseline diffusion model's FID is still larger than 3.0 with even 18 NFEs. Since both models have the same cost for each NFE, we conclude that the DPDM is about 3 times more efficient than diffusion models for few-step generations.
>
> However, as we pointed out in Section 6, the performance of DPDM with sufficient NFEs (FID of 2.68 with 8 NFEs) is still worse than the diffusion model (FID of 1.97 with 35 NFEs). This is currently a limitation of DPDM and we are very glad to further explore the performance ceiling of DPDM through advanced training configurations, tricks, and other aspects.
>
> **Q3**. What does $\tilde{x}$ stand for in Equation 1? I don't think it has been defined anywhere.
>
> **A3**. We appreciate your careful reading and We feel sorry for the confusion. There is a typo in Equation 1 in Algorithm 1. Here the $\tilde{x}_t$ is a mistake for $\hat{x}_t$ in the expectation operation. Thank you for pointing out the typo.
>
> We will incorporate your suggestions on other minor issues in the revision. We hope our answers have resolved your concerns. If you still have concerns, please let us know.

---

### Official Review · Reviewer_u1HV · 2023-11-09

**Soundness:** 2 fair
**Presentation:** 3 good
**Contribution:** 2 fair
**Rating:** 5
**Confidence:** 3

**Summary:**

This paper studies student-teacher fine-tunning method that improves and accelerates sampling steps for Diffusion Models (DM). The primary hypothesis motivating this research asserts that the conventional score matching objective used in training leads to suboptimal denoisers for DM, thereby limiting the generation of high-quality samples when constrained to a minimal number of sampling steps (NFE < 10). Results are shown on two datasets comparing to several most recent  baselines.

**Strengths:**

1, The beginning of the article (Section 1 and Section 2) is well written and presents the motivation behind the introduction of the proposed student-teacher fine-tunning method in a very didactic way.

2,  There are some theoretical justifications for the training gradient of the proposed smoothed KL.

3,  Numerical experiments on small datasets (e.g., 64X64) verify that the proposed DPDM outperforms previous sampling methods in terms of quality (FID) with fewer function evaluations

**Weaknesses:**

1, While Diff-Instruct was not explicitly designed to accelerate diffusion model sampling, the proposed DPDM still shares many similarities in format, such as the smoothed KL divergence, student-teacher fine-tuning, and the gradient update of the 'objective.' More importantly, Diff-Instruct tested its sampling quality with small NFEs as well. These aspects make the actual novelty of DPDM not clear to the reviewer.

2, While there are some theoretical demonstrations concerning the gradient of KL divergence for a single denoiser, equivalent analyses for multiple denoisers in Eq.3.4 appear to be missing, making the work somehow incomplete.

3,  Compared to other training-free sampling acceleration methods, the computation costs of DPDM are still relatively heavy, making the practical impacts unclear.

4, This work is only demonstrated for image synthesis on small datasets, while other conditional sampling tasks (e.g., text-to-image or image-to-image translation) are missing.

**Questions:**

1, What is the runtime comparison between the proposed and other methods ?

2, Can this strategy be directly adapted to other conditional sampling methods, such as image-to-image translation? On the other hand, given that the sampling trajectory is implemented by different denoising operators, how robust is the DPDM to variations at intermediate stages, especially at low noise levels ?

3, In Table 4, it seems that DPDM performs worse than Diff-Instruct for NFE < 4. What accounts for such differences?

4, When training multiple denoisers, how does convergence occur in those student networks?

---

> ### Author Response · Authors · 2023-11-14
> **Thank you for your useful feedback (1)**
>
> Thank you for your useful feedback, we will incorporate them in the revision.  We will address your concerns one by one in the following paragraphs. Before that, we first give a summary of the main contributions of our work.
>
> In this work, we propose a novel stand-alone generative model termed the Data-prediction denoising model (DPDM), which achieves significantly better generative performance than diffusion models under the few-step sampling settings.
>
> **Q1**. DPDM shares similarities with Diff-Instruct [1], which also performs well with few sampling steps. Could you give some more discussions?
>
> **A1**. We highly appreciate your keen intuition. In this work, when defining the DPDM, we draw inspiration from two amazing works: the diffusion models (i.e., the EDM work [2]) and the Diff-Instruct [1] (which achieves strong performance for single-step generation). The DPDM combines the advantages of both worlds, resulting in strong few-step generative performances. However, such a combination of both works is non-trivial, because:
>
> (1). Though the Diff-Instruct achieves state-of-the-art single-step diffusion distillation performance during its publishing, it does not support multi-step sampling, because the Diff-Instruct can only train a one-step student model to directly transport a Gaussian latent vector to obtain generated samples. The most amazing part of Diff-Instruct is its mathematical foundations (i.e. the formulation of divergence minimization when deriving the algorithm). However, the possibility (with improved performances) of incorporating such a math formulation for stronger few-step generative modeling has not been yet explored. To our knowledge, we are the first work that explored and successfully achieved a strong few-step generative performance with an FID of 3.66 on ImageNet$32\times 32$ data with 8 sampling steps, which is stronger than Diff-Instruct (which only supports one-step sampling) and baseline diffusion models.
>
> (2). Though diffusion models can generate promising data samples when the sampling steps are sufficient, they perform poorly under a few-step setting (even with the best solvers). In this work, we analyze and locate the reason for such a performance drop under a few-step setting as the weak distribution recovery ability of diffusion models' denoiser (Section 5.1 in the submission). Such an observation of weak denoisers not only motivates us to propose DPDM with improved denoisers but also sheds light on other future work which potentially aims to address the weakness of diffusion models.
>
> (3). We incorporate all enhanced denoisers through a single $\mathbf{d}_\theta(x_t,t)$ network with weight sharing. This overcomes the heavy computational and memory cost when storing multiple denoisers.
>
> **Q2**. Put more discussions on Eq.3.4.
>
> **A2**. We feel sorry for the confusion. Equation 3.4 is a generalization of Equation 3.3 which only considers a single noise level. For a single noise level, we only consider a single denoiser. When generalizing to multiple denoisers, we use a weight-sharing multi-level denoiser network $\mathbf{d}_ \theta(x_t,t)$ to parametrize multiple denoisers in a single network $\mathbf{d}_\theta$.
>
> Therefore, the gradient formula to update $\theta$ of $\mathbf{d}_\theta(x_t,t)$ is a weighted combination of the single-step gradient Equation 3.3 through a weighting function $w(t)$ over diffusion time t (or over different noise levels). We really thank you for your careful reading and we acknowledge that there is a small typo in Equation 3.4.:
>
> the denoiser should write $x_t = \mathbf{d}_ \theta(x_0 + \sigma(t)\epsilon, t) + \tilde{\epsilon}$  instead of $x_t = \mathbf{d}_ \theta(x_0 + \epsilon) + \tilde{\epsilon}$, and the Equation 3.4 should be modified to
>
> $$ \operatorname{Grad}(\theta)= \int_ 0^ T w(t)\mathbb{E}_ { \boldsymbol{x}_ t=\boldsymbol{d}_ \theta(\boldsymbol{x}_ 0 + \sigma(t)\epsilon, t)+\tilde{\epsilon} \atop \boldsymbol{x}_ 0 \sim p_ 0, \epsilon,\tilde{\epsilon} \sim \mathcal{N}(\boldsymbol{0}, \boldsymbol{I})} \big[\boldsymbol{s}_ {q_t}( \boldsymbol{x}_ t, t) - \boldsymbol{s}_ { p_ t}( \boldsymbol{x}_ t, t) \big] \frac{\partial \boldsymbol{x}_ t}{ \partial \theta} \mathrm{d}t. $$
>
> We hope fixing this typo will make our presentation more clear and resolve your concerns.

---

> > ### Author Response · Authors · 2023-11-14
> > **Thank you for your useful feedback (2)**
> >
> > **Q3**. Compared to other training-free sampling acceleration methods, the computation costs of DPDM are still relatively heavy, making the practical impacts unclear.
> >
> > **A3**. We are sorry for the confusion. As we have noted in A1 and A2, we use the same neural architecture as the diffusion model (i.e. a network that takes both input noisy data and a time index), therefore, the inference cost of DPDM per step is exactly the same as that of the diffusion model (and related sampling solvers). Therefore, the inference cost per step of DPDM is the same as the diffusion model. However, the DPDM achieves significantly better performances than diffusion models with training-free solvers. As a comparison, on the CIFAR10 dataset, the DPDM achieves an FID of 3.85 with 5 NFEs, while the best-performing SA-Solver has an FID of 4.91, which indicates that the DPDM is at least 3 times more efficient when inference than training-free solvers.
> >
> > As for the training costs, since the DPDM is a stand-alone generative model, the training cost is inevitable. But in this work, our goal is not only **accelerate** diffusion model sampling but also to explore the possibility of a **new approach to generative modeling** that can outperform diffusion models under fewer-step generative settings.
> >
> > **Q4**. What is the runtime comparison between the proposed and other methods?
> >
> > **A4**. The comparison to training-free diffusion solvers is presented in A3. Here we take another multi-step generative model, the consistency model [3], as a baseline and present a comparison of DPDM with CM. Both the DPDM, CD, and baseline EDM models share the same neural architecture, therefore, their inference cost per step is exactly the same. **Table 6** in the Appendix of the submission compares the memory costs of DPDM and CD, which shows that the DPDP's training consumes slightly more (less than 10 percent) memory than CD.
> >
> > **Q5**. Can this strategy be directly adapted to other conditional sampling methods, such as image-to-image translation? how robust is the DPDM to variations at intermediate stages, especially at low noise levels?
> >
> > **A5**. As in A1 and A2, we point out that the DPDM shares the same neural architecture as a diffusion model (weight-sharing denoises). For other applications, such as i2i, there are no technical challenges to using DPDM, because the experiences on both the architecture and the training paradigm can be transferred with minor modifications on loss functions and inference pipelines from a huge amount of existing work that uses diffusion models for i2i. Since we use only a single network instead of multiple separated networks, we do not think the model will become non-robust on i2i tasks.
> >
> > **Q6**. In Table 4, it seems that DPDM performs worse than Diff-Instruct for NFE < 4. What accounts for such differences?
> >
> > **A6**. We appreciate your keen intuition. We acknowledge that currently, the DPDM performs worse than Diff-Instruct for single-step generation. But it is a lot stronger than Diff-Instruct under few-step settings. Besides, we do think more efforts, innovations, and **tricks** may help train stronger denoisers for DPDM, such as incorporating more auxiliary models  (similar to StyleGan-xl with eight auxiliary discriminators), more auxiliary loss functions, etc. These orthogonal extensions will complement the DPDM, resulting in stronger performance. However, our main goal is to explore the possibility of a new strong few-step generative model by addressing the weaknesses of diffusion models. Therefore, we deliver a minimal implementation with no extensions in order to demonstrate clearly the differences between DPDM and DM and highlight the advantages of DPDM.
> >
> > **Q7**. When training multiple denoisers, how does convergence occur in those student networks?
> >
> > **A7**. As we point out in A1 and A2, we use a single network to parametrize multiple denoisers. Therefore, there is no concern about balancing the convergence of multiple denoiser networks. Besides, we show in theory that algorithm 1 leads to a convergence of smoothed KL divergence between DPDM's denoiser distributions and the data distribution. Additionally, our empirical studies (Section 5.2) have shown that the training algorithm is efficient yet stable.
> >
> > In conclusion, we thank you for your useful suggestions. We hope our answers have resolved your concerns, and if you still have any concerns, please let us know, and we will be glad to further address them.
> >
> > [1] Diff-Instruct A Universal Approach for Transferring Knowledge From Pre-trained Diffusion Models
> >
> > [2] Elucidating the Design Space of Diffusion-Based Generative Models
> >
> > [3] Consistency Models

---

> ### Author Response · Authors · 2023-11-20
> **Thank you for your reviews! we are glad to provide more clarifications.**
>
> Dear reviewer:
>
> Thank you for your reviews! The author-reviewer rebuttal period is coming to a close soon. We sincerely hope that our responses have adequately addressed the concerns you have raised.
>
> In this work, we introduce a **new class of multi-step generative models**, named Data-prediction Denoising Models, which show significantly **stronger performance than diffusion models** under the few-step setting. We also explore the weak data distribution recovery ability of the diffusion model's denoisers and practical solutions for enhancing them. We believe that our findings and solutions may benefit future researchers in either improving diffusion models or proposing new generative models.
>
> We hope our rebuttal has resolved your questions. If you still have any unresolved concerns or additional questions, please do let us know! We would be very glad to provide more clarification and address any remaining issues.
>
> Best wishes,
>
> Authors of the submission #5134.

---

> > ### Comment · Reviewer_u1HV · 2023-11-22
> > **Thank You for Addressing My Comments**
> >
> > The reviewer would like to thank the authors for taking the time to address their comments. After reviewing the authors' responses, I find this work to be interesting and novel enough for publication. However, there is still a lack of strong evidence supporting the direct application of this distillation technique to large-scale image/video generation. As a result, the reviewer must maintain their original rating.

---

> > > ### Author Response · Authors · 2023-11-23
> > > **Thank you for appreciating our work!**
> > >
> > > Thank you for appreciating our work! We are glad that you highlighted the novelty of our proposed DPDM model. We acknowledge that large-scale experiments will make the submission a very strong acceptance.
> > >
> > > * First, we would like to say, that the main goal of this work is to explore practical methods for developing new multi-step generative models, therefore, we choose to compare the DPDM with other models on CIFAR10 and ImageNet64 which requires minimal additional implementation techniques which may influence the fairness of the comparison of each model (other than tricks) and multi-step samplers.
> > >
> > > * Second, the CIFAR10 and ImageNet64 datasets are the most explored image generation benchmarks, on which nearly all other methods have shown their performances. Therefore, it is suitable and sufficient to have a comprehensive understanding of DPDM on them. Other works, such as EDM, Diff-Instruct, and One-step DEQ, etc., have all conducted comparisons on CIFAR10 and ImageNet64 datasets for comparisons.
> > >
> > > * Third, we think the scaling-up work of DPDM to larger scales, such as the popular text-to-image generation and image-to-image translation applications requires a significant amount of computational resources, engineering implementation, hyper-parameter tuning, and tricks. Frankly speaking, such requirements are often related more to the engineering implementation techniques rather than the generative modeling methodology itself. Therefore, we believe such additional engineering implementations deserve new work in the future.
> > >
> > > However, we highly value your constructive suggestions and we are elaborately trying large-scale dataset experiments, but it may take some time because we do not have sufficient GPU devices;
> > >
> > > Best regards,
> > >
> > > Authors of the submission #5134.

---

### Author Response · Authors · 2023-11-23
**Paper revision**

## reminder for revision

We have made significant updates to our draft to improve the overall quality of our submission. We mark the changed part with the color of the orange. Here are the major changes:

* We add a special paragraph in the introduction section, following the constructive suggestions from the Reviewer twcE, to highlight the differences between our DPDM and traditional diffusion models in both the training method and the sampling algorithms. **This revision greatly improves the presentation of DPDM to prevent confusion about DPDM and traditional DDPM models**;

* We add a short paragraph in Section 3.1 to understand the reason for the weak denoiser of diffusion models by pointing out that the DM's denoisers are naturally an averaged data sample. **This revision provides stronger motivation and insight for DPDM and tackles a key weakness of diffusion models**;

* We address the typo in Algorithm 1 and Equation (3.3), following the feedback of the Reviewer u1HV. **This revision improves the presentation and accuracy of the training algorithm of DPDM**.

* We address some other typos raised by the Reviewer X9iJ and ourselves.

We are grateful for the encouraging feedback from all reviewers and would like to thank all reviewers sincerely. We have and will take all suggestions into account carefully to enhance the quality of our paper. We appreciate all reviewers' time and effort in providing us with your valuable insights.

Best regards,

Authors of the submission #5134.

---

### Meta-Review · Area_Chair_1yhK · 2023-12-06

**Metareview:**

The paper proposes Data-Prediction Denoising Model (DPDM), a class of generative models very similar to diffusion models. The reviewers acknowledge that DPDMs performs well on small benchmark data, however the reviewer also agree that experimental results on high resolution data would be very useful:

* Reviewer u1HV (5) finds a few weaknesses, and after the rebuttal and discussion the weakness that remains is a lack of evidence supporting the application of the proposed technique to large-scale image or video generation, and thus is not supportive of publication.

* Reviewer X9iJ (6) also notes that the scope of the experiments is limited, and that even though the focus of the work is on inference efficiency, inference metrics are not provided. The paper clarified in what sense the method is efficient.

* Reviewer twcT (5) identified weaknesses regarding the theoretical analysis and exposition that could be cleared up for the most part.

* Reviewer gGZJ (8) finds the paper addresses a useful and practical issue with diffusion models, and is novel and well structured. The reviewer is also concerned on whether the paper solves a practical problem since it requires multiple components, and that the paper focuses predominantly on low-resolution generation tasks.

The paper looks very promising, however, at this point the paper is somewhat unfinished: code to reproduce the results is missing, there were multiple issues regarding the presentation in the original manuscript, and currently the paper is (slightly) too long. Results on higher resolution data would significantly strengthen the claims; I understand the authors reasoning that they do not have the compute to train a stable-diffusion-scale model, but this is not necessary, the paper can be extended significantly with higher-resolution data and experiments that do not require huge amounts of compute.

**Justification For Why Not Higher Score:**

The paper is somewhat unfinished, larger scale experiments and code is missing.

**Justification For Why Not Lower Score:**

N/A

---

### Decision · Program_Chairs · 2024-01-16

Reject